

# Enhancing cybersecurity through autonomous knowledge graph construction by integrating heterogeneous data sources

Hatoon Alharbi[1], Ali Hur[2], Hasan Alkahtani[1] and
Hafiz Farooq Ahmad[1]

[1] Computer Science Department, College of Computer Sciences and Information Technology
  (CCSIT), King Faisal University, Eastern Province, Al-Ahsa, Saudi Arabia
[2] School of Science, Edith Cowan University, Joondalup, Western Australia, Australia

## ABSTRACT

Cybersecurity plays a critical role in today's modern human society, and leveraging knowledge graphs can enhance cybersecurity and privacy in the cyberspace. By harnessing the heterogeneous and vast amount of information on potential attacks, organizations can improve their ability to proactively detect and mitigate any threat or damage to their online valuable resources. Integrating critical cyberattack information into a knowledge graph offers a significant boost to cybersecurity, safeguarding cyberspace from malicious activities. This information can be obtained from structured and unstructured data, with a particular focus on extracting valuable insights from unstructured text through natural language processing (NLP). By storing a wide range of cyber threat information in a semantic triples form which machines can interpret autonomously, cybersecurity experts gain improved visibility and are better equipped to identify and address cyber threats. However, constructing an efficient knowledge graph poses challenges. In our research, we construct a cybersecurity knowledge graph (CKG) autonomously using heterogeneous data sources. We further enhance the CKG by applying logical rules and employing graph analytic algorithms. To evaluate the effectiveness of our proposed CKG, we formulate a set of queries as questions to validate the logical rules. Ultimately, the CKG empowers experts to efficiently analyze data and gain comprehensive understanding of cyberattacks, thereby help minimize potential attack vectors.

## INTRODUCTION

Cybersecurity is a rapidly developing domain due to the increasing number of cyberattacks (*Saravanan & Bama, 2019*). Since the Internet is integrated into every aspect of the lives of people, it is a vast network where it has become a place for cybercriminals by using its technologies to attack individuals and corporations. These attacks can damage vital resources that lead to serious harm to the assets of any organization or even an individual's personal information (*Uma & Padmavathi, 2013*). Both small and large organizations, regardless of their size, and sector, face various cybersecurity issues on daily basis. Confidential information must be protected from the concern at different layers of the

Corresponding author
Hatoon Alharbi,
kiaral1919@gmail.com

cyberspace. Recently, there are different attacks that are developed swiftly, which exploit different types of vulnerabilities (*Abomhara & Køien, 2015*). These attacks pose a serious threat to Internet security. Knowledge of threats and vulnerabilities helps security analysts maintain improved security and successfully stop attackers from using flaws to launch attacks (*Sun et al., 2020*). When security analysts deeply analyze vulnerability information, discovering the implied relationship between relevant information has high importance to resist external attacks and discover and repair vulnerabilities at the right time (*Wang et al., 2020*). Therefore, acquiring and managing a significant number of high-quality vulnerability data is very important in cybersecurity domain (*Qin & Chow, 2019*).

Different sources of vulnerability datasets exist such as open-source vulnerability database (OSVDB) (*osv.dev, 2021*), Symantec/Security Focus BID database (*Symantec, 2017*), and National Vulnerability Database (NVD). NVD is a well-known vulnerability database that is maintained by the National Institute of Standards and Technology in the USA (*National Vulerability Database (NVD), 1997*). NVD connects chain of datasets related to vulnerabilities, which include common vulnerabilities and exposures (CVE), common platform enumeration (CPE), common vulnerability scoring system (CVSS), and common weakness enumeration (CWE). Cybersecurity analysts need up-to-date information related to cybersecurity. As mentioned above, cybersecurity information is obtained from different sources and formats, which makes it very complex to sort out relevant and irrelevant information. As the number of relevant information has been growing fast, it is difficult to manage and use this information, therefore, this information is needed to be unified and organized. Different sources of information are required to be collected, integrated, and linked together to analyze vulnerabilities comprehensively. Consequently, it is important to develop a unified knowledge representation that integrates all information from different sources and formats to enable cybersecurity analysts to have improved visibility, awareness, and in-depth analysis (*Du et al., 2018*). However, this task requires a lot of effort and increases the security analysts' workload (*Sun et al., 2020*). A knowledge graph (KG) can be used to represent this information related to cybersecurity. KG represents information in the form of concepts, entities, and relationships between entities, which makes the data more understandable and readable in a way that it becomes machine-processable and interpretable. Given a knowledge graph KG = <E1, R, E2>, where E1, E2 represent the set of entities, R represents the relation. For instance, a triple contains (attack, exploits, vulnerability) where an attack is represented as a subject, an exploit as a predicate, and a vulnerability as an object. As illustrated in Fig. 1, an attack is any illegal or malicious activity that exploits a vulnerability to harm the assets of any organization or individual. KGs first appeared in the 1990s as a result of progress in the information extraction domain (IE) (*Kejriwal, 2022*). A KG is used to merge the structural information of concepts from multiple information sources and links them together (*Kim et al., 2020*). KG is beneficial for many different fields especially cybersecurity field (*Agrawal et al., 2022*). For example, KGs are used to enhance search result ranking in Google search engine (*Ilievski, Szekely & Zhang, 2021*).

Although KGs are highly useful in the cybersecurity field to organize, manage, and employ a large amount of information effectively, the task of constructing KGs is highly

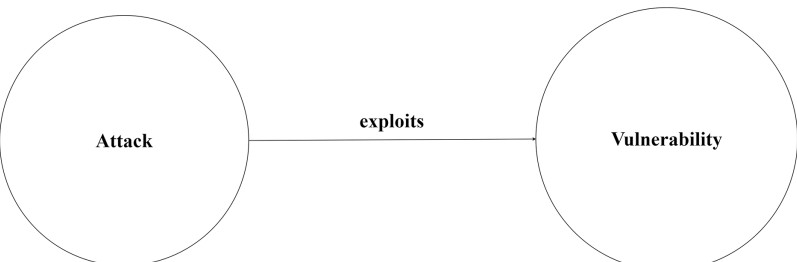

**Figure 1** **Basic building block of the knowledge graph.**

challenging. Whether the process of construction is automated or done manually, it is expensive, error-prone, and requires a lot of human effort. Moreover, it is important to construct graphs using both structured and unstructured data sources. However, the use of structured and unstructured data is more challenging since the integration of data is more difficult to achieve (*Masoud et al., 2021*). For example, the ambiguity and imprecision of natural language make the automatic construction of a KG is very challenging. In addition, KGs may have difficulty to interpret natural language queries or contexts because of their KGs may have difficulty interpreting natural language queries or contexts because traditional KGs are unable to understand context, word ambiguity, or implicit meanings in the context.

Additionally, the complexity of unstructured data makes data processing a challenging task as it requires sophisticated techniques and algorithms (*Paulheim, 2016*). Also, the size and diversity of KGs that are constructed automatically make accuracy a challenging problem (*Ojha & Talukdar, 2016*). KGs usually have some shortages in the current knowledge as it contains data insufficiency, redundant information, inconsistencies, and incomplete data (*Liu et al., 2022*), which lack coverage of the CKG. Thus, there is a need to use KG reasoning methods to improve the KG, in a way to draw conclusions from known facts to infer new knowledge such as logical rules (*Fang, Qi & Yue, 2020*). In addition, use of information from different sources and formats to improve the coverage of KGs implies the need to handle structured data as well as unstructured data. The integration of information extracted from structured and unstructured data improves coverage of knowledge graphs (*Kejriwal, 2022*).

In this article, we construct a knowledge graph autonomously labelled property graphs (LPGs) (*Robinson, Webber & Eifrem, 2015*). Our proposed using framework consists of three main parts natural language processing (NLP) part, a knowledge graph construction, and logical rules to refine and improve KG. We use information extraction (*Zhao, Pan & Yang, 2020*) and a set of NLP techniques (*Kumar & Manocha, 2015*) to construct a knowledge graph from unstructured textual sources. Our KG consists of information obtained from structured and unstructured data (*i.e.*, heterogeneous data). We also use logical rules to improve the coverage of this knowledge graph. Additionally, we apply graph analytics algorithms to gain more information about KG. In this research, we answer the following question: How can we improve KG construction using logical rules?

The main contributions of this article are:

- Autonomous knowledge graph construction from structured and unstructured sources based on the LPGs.
- Design and implementation of the model for logical rules to improve KG.
- Utilize diverse graph analytics algorithms to evaluate performance of our KG.

## BACKGROUND OF CYBERSECURITY KNOWLEDGE GRAPHS

The cybersecurity knowledge graph (CKG) is a type of KG that is related to the cybersecurity domain. CKG contains nodes (*i.e.*, entities) and edges (*i.e.*, relationships) that comprise a comprehensive security semantic network which have different attacks and defense scenarios (*Qin & Liao, 2022*). In a CKG, nodes represent entities such as the name of vulnerability and the pattern of attack, and edges represent the relationships between entities. Researchers have conducted a detailed comparison of works related to CKG and provided a detailed systematic literature review on information extraction from unstructured data (*i.e.*, CTI reports) (*Zhao et al., 2022*). Various studies constructed different types of CKG from different perspectives of cybersecurity such as security assessment and attack investigation (*Han et al., 2018*; *Kiesling et al., 2019*). A knowledge graph is highly useful in the cybersecurity field to organize, manage, and use different cybersecurity-related information. The knowledge graph can extract and combine existing knowledge from heterogeneous data from different sources in an effective manner, by using KG construction and refining methods such as ontology (*Rastogi et al., 2020*), and information extraction (*Husari et al., 2017*; *Zhao et al., 2020*). In addition, KGs are very efficient since they can express the knowledge in the cybersecurity field in a relational and structural manner, and visualize the knowledge graphically. Moreover, CKGs can simulate the security specialists' thinking process aims to derive new relations or verify data consistency based on the existing facts (triples) and logic rules (*Ji et al., 2022*), this can be done by using semantic modelling, query, and reasoning methods.

Generally, the process of constructing a cybersecurity knowledge graph (CKG) includes data sources, information extraction techniques, and cybersecurity ontology. Cybersecurity ontologies are divided into two main categories: unified security ontologies such as UCO, which provide a broad framework for categories: unified security ontologies such as UCO, which provide a broad framework for cybersecurity concepts, and specific scenario ontologies such as vulnerability analysis ontologies, which focus on specific cybersecurity domains. The choice of ontology depends on the particular problem being addressed. Cybersecurity ontologies are used to formally represent cybersecurity concepts and their relationships, aiding in knowledge organization and reasoning (*Liu et al., 2022*). On the other hand, unstructured data, such as text documents or social media posts, often contains valuable information that requires advanced techniques like open information extraction (OIE) (*Gashteovski et al., 2021*; *Owen & Widyantoro, 2017*), named entity recognition (NER), or relation extraction (RE) (*Li et al., 2022*) to be extracted and

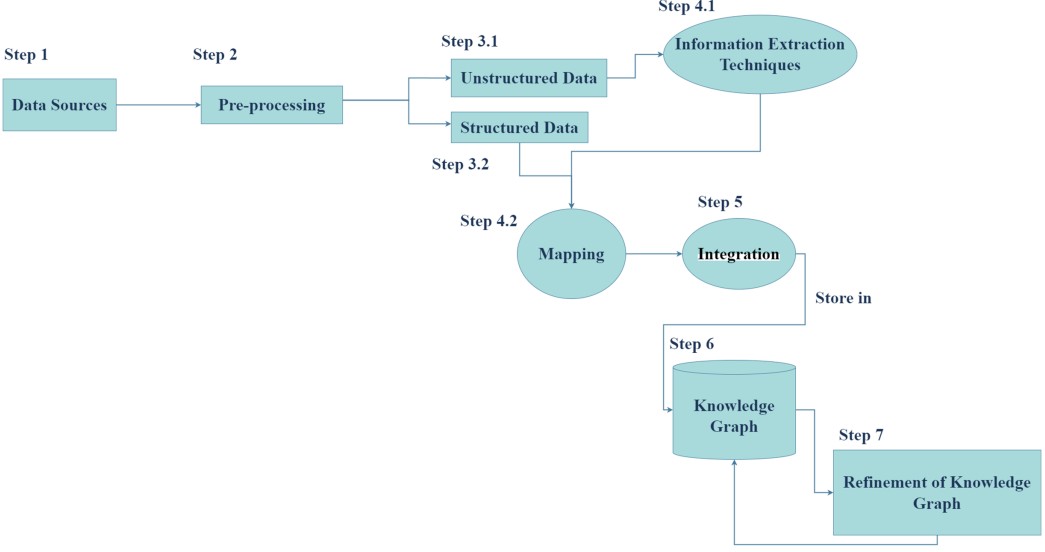

**Figure 2  Knowledge graph construction process.**

analyzed. By combining the structured knowledge from ontologies with the extracted information from unstructured data, we can gain deeper insights into complex cybersecurity challenges and develop more effective solutions. So, we followed the knowledge graph construction process to propose our framework, is demonstrated in Fig. 2.

# RELATED WORK

The current data sources in the cybersecurity field are semi-structured and unstructured data that are difficult to machine-understand and reuse. The majority of the previous work concentrated on providing knowledge graphs, that used different information extraction techniques on unstructured data (*Jia et al., 2018*; *Kim et al., 2020*; *Qin & Chow, 2019*; *Sarhan & Spruit, 2021*; *Sun et al., 2020*, *2023*). In contrast, some of the earlier works concentrated on providing a knowledge base (*Jia et al., 2018*).

## Information extraction techniques

In NLP, information extraction techniques are used to automatically extract structured information from unstructured text data. Several studies either have implemented or proposed NER to extract entities from unstructured text (*Jia et al., 2018*; *Kim et al., 2020*; *Qin & Chow, 2019*; *Sarhan & Spruit, 2021*; *Sun et al., 2020*, *2023*).

*Sarhan & Spruit (2021)* proposed an Open Cyber Threat Intelligence (CTI) Knowledge Graph (Open-CyKG) framework to extract useful cyber threat information from unstructured advanced persistent threat (APT) reports by using an attention-based neural OIE model. They developed neural cybersecurity NER that aids in labeling the relation triples created by the OIE model. By developing this NER they were able to identify the relevant entities. Then, using the fusion techniques and word embeddings, the extracted structured data is canonicalized to construct the KG. The findings demonstrated that their NER model achieved over 95% F-measure.

*Sun et al. (2020)* proposed a vulnerability knowledge graph by integrating heterogeneous data. They constructed a vulnerability ontology *via* analyzing the CNNVD (China National Vulnerability Database of Information Security), NVD, CWE, CVSS, Common Attack Pattern Enumeration and Classification (CAPEC) with the other heterogeneous data. Then, they used a rule-based entity recognition method for extracting vendor names from the unstructured vulnerability description text. The results illustrated that the accuracy of the entity recognition method achieved 89.76%, and the results can achieve increase in the accuracy with the decrease in the error rate if more rules being used in entity recognition.

*Jia et al. (2018)* suggested a quintuple cybersecurity knowledge base model composed of concepts, instances, relations, properties, and rules. They adopted a method that combined rule-based and machine learning. To construct a cybersecurity knowledge base, they extracted entities and built ontology using machine learning. Then, by calculating formulas and using the path-ranking algorithm, new rules were derived. To extract useful information, an extractor can be trained by using Stanford-NER. Findings illustrated that the NER is capable of providing several features, and the use Gazettes parameter could be used to train a recorganizer in the cybersecurity field. In the absence of a gazette, recognition results of NER approximately achieved 73% precision, recall, and F1.

*Kim et al. (2020)* suggested an approach that extracts key information from cyber threat intelligence (CTI) reports automatically utilizing a NER system. The authors specified the meaningful keywords in the security field as entities such as URL and IP address, *etc*. Then they connected the words extracted from the text data of the report with these keywords. Additionally, they used a conditional random field network to use the character-level feature vector as an input to bidirectional long-short-term memory to achieve higher performance. The findings showed their model achieved an average F1 score of 75.05% with a standard deviation of 1.74.

*Qin & Chow (2019)* proposed KG called VulKG from vulnerability databases including CVE, CWE, and CVSS, however, but it did not include details on information regarding CWE and CAPEC. The researchers suggested an automatic analysis and reasoning model based on the vulnerability knowledge graph. Then, this model used to analyze the vulnerability descriptions and extract named entities and include these entities into KG. They also used knowledge graph completion technique to discover the hidden relationships between weaknesses in the graph, which is based on chain confidence. The results demonstrated that the AARV model accuracy of the technique achieved 82.69%. However, the example could simply somewhat take over the place of the security specialists' analysis and labeling work under certain scenarios, as the operator is required to know the query target in advance. Also, querying and re-using data are complicated if the data are considered in association with privacy policies.

*Sun et al. (2023)* proposed SecTKG, which is an automated architecture for constructing knowledge graphs for open-source security tools. An ontology model was created to characterize the properties of security tools, their users, and their associations within the open-source community. A knowledge graph based on the security tools' ontology can be applied in practice using some graph algorithms. As a result, they developed a method for

measuring the influence of security tools using multiattribute decision-making and linear threshold propagation. In order to find some security tools that are high-quality, well maintained, but less famous, this method examines node attribute information, different types of relationships, and social network analysis.

While some studies have proposed relation extraction techniques to extract relationships (*Jones et al., 2015*; *Pingle et al., 2019*; *Piplai et al., 2020*; *Tovarňák, Sadlek & Čeleda, 2021*). *Pingle et al. (2019)* proposed the RelExt system to create semantic triples on cybersecurity text. By extracting possible relationships they used deep learning techniques. The cybersecurity knowledge graph can be confirmed using the semantic triples set generated by their system. This system can extract the relationships between two named entities. RelExt takes a pair of named entities extracted from a cybersecurity text as inputs. A set of entity relationships is output by RelExt. Based on the schema of the cybersecurity knowledge graph, the input of RelExt can be further processed. Therefore, the inconsistent entity pairs with UCO 2.0 and STIX 2.0 were removed. The findings illustrated that the RelExt system predicted successfully more than 700 relationships from malware descriptions (Dark Caracal and CrossRat). Based on various data splits, the accuracy achieved above 95%.

*Piplai et al. (2020)* proposed a pipeline to extract information from after action reports (AARs), aggregate the extracted information by fusing similar entities, and represent that extracted information in a CKG. They built a customized named entity recognizer called 'Malware Entity Extractor' (MEE) to extract entities. Then, they created a neural network for predicting how 'malware entities' pairs were related to each other. When the authors predicted entity pairs and the relationship between them, they asserted the 'entity-relationship set' in a CKG. For enhancing the CKG, they fuse similar entities, this fusion assists represent intelligence extracted from multiple documents and reports.

*Jones et al. (2015)* suggested that the bootstrapping algorithm can provide automated information extraction targeting security documentation, to assist security professionals to find and understand information on vulnerabilities, attacks, or patches on their network. This algorithm is used to extract security entities and their relationships from the text. The bootstrapping algorithm requires a few relations or patterns as inputs, that contained an active learning component to prevent drifting from the desired relations by queries the user on the most vital decisions. The results illustrated that the precision of the tested small *corpus* achieved around 82%.

In addition to the above mentioned studies, *Rahman et al. (2024)* proposed ChronoCTI which is an automated NLP and ML-based pipeline to extract temporal attack patterns from cyberthreat intelligence (CTI) reports of previous cyberattacks. The ChronoCTI construction process begins with building the temporal attack patterns ground truth dataset, and performing advanced large language models, natural language processing, and machine learning techniques. The results showed that ChronoCTI performance was good in precision however it lacks recall. By applying ChronoCTI on more than 700 CTI reports, they have determined 124 temporal patterns that are categorized into nine pattern categories.

*Høst (2022)* constructed a vulnerability knowledge graph using common weakness enumeration (CVE) records from the National Vulnerability Database (NVD). His proposed architecture consists of data processing and labeling, named entity recognition and relation extraction, and entity prediction. The NER model was used for extracting cyber-security entities to train on labeled NVD data. Then, for extracting relations between entities, a rule-based relation extraction (RE) model was used. The RE model determined the standard structure of CVE records since it's based on NVD labels. Triples were used to form their initial knowledge graph, these triples were the result of combining the extracted entities and their relations. A knowledge graph embedding method (TuckER) was used to predict missing entities to enhance the initial graph. The results showed that the TuckER method achieved 0.76 in the Hits@10 score. However, the data consists of all unique CVE IDs, which can be difficult for the model to predict due to the granularity level related to the limited discriminatory information. In addition, many overlapping vulnerable versions of different products can make the prediction process challenging in their model. The following Table 1 provides literature review summary.

## Types of data source

KGs are constructed using various types of data sources, including structured, semi-structured, and unstructured data. The majority of studies in the literature leverage information extraction techniques to process and derive insights from unstructured data sources, such as Advanced Persistent Threat (APT) reports, After-Action Reports (AAR), and CTI documents. However, a subset of research has incorporated heterogeneous data sources into their KGs, as demonstrated in the works of *Jia et al. (2018)*, *Qin & Chow (2019)*, and *Sun et al. (2020)*. Consistent with the approach of *Sun et al. (2020)*, this study focuses on specific fields within CWE and CVE documents, including descriptions and CPE, among others.

## Graph-based analytics

Graph analytics algorithms are used to determine patterns and relationships among objects. There are different types of graph analytics algorithms such as centrality and community detection algorithms. These algorithms are used for graph analytics in Neo4j. Graph analytics provides comprehensive and detailed statistics on graph databases to reveal hidden meaning to drive discovery of new information (*Needham & Hodler, 2018*).

Centrality algorithms are one of the most important algorithms. These algorithms determine the important nodes in a given graph. The importance of the node indicates that this node has many direct connections, this node is transitively connected to other important nodes and reaches other nodes with some hops. There are different types of centrality algorithms such as Degree and PageRank algorithms. The Degree algorithm is employed to determine the popular nodes inside the graph. It calculates the incoming relationships, outgoing, or both from a node, based on the relationship projection direction. While the PageRank algorithm is applied to calculate the node's importance depending on the incoming relationships and the corresponding source nodes' importance. Whereas community detection algorithms are employed to assess how groups

**Table 1 Literature review summary.**

| Author | Source | Nature | Volume of data points | Fields |
|---|---|---|---|---|
| Sarhan & Spruit (2021) | Microsoft Security Bulletins dataset, Malware-specific dataset (collected from various CTI reports). | Unstructured | 5,072 sentences (Microsoft Security Bulletins dataset) 3,450 sentences (Malware-specific dataset). | – |
| Sun et al. (2020) | NVD, CNNVD, CWE, CVSS, CAPEC. | Both | Not specified | Vendor, product, vulnerability, weakness, attack pattern, Result. |
| Jia et al. (2018) | Cybersecurity datasets (e.g., SNORT, chinabaike, Adobe, Microsoft). | Both | Not specified | Vulnerability, Assets (Software, OS), Attack. |
| Kim et al. (2020) | CTI reports | Unstructured | 160 unstructured PDF documents | – |
| Qin & Chow (2019) | NVD | Both | Not specified | CVE, CWE, CPE, CVSS information. |
| Sun et al. (2023) | The open-source tools' dataset obtained from GitHub. | Unstructured | Above 40 thousand tool repositories' information, above 3 million users' information. | – |
| Pingle et al. (2019) | Technical reports, blogs, CVE, Microsoft and Adobe security bulletins, STIX corpus. | Unstructured | 474 detailed technical reports and blogs, nearly 90,000 JSON entities (CVE). | – |
| Piplai et al. (2020) | CVE, NVD, STIX, Microsoft security, Adobe security, annotated AARs, AARs. | Both | About 3,600 sentences (MEE evaluation). 90,000 annotated relationships, triples, annotated triple (RelExt evaluation). | Software, Exploit-Target, Malware, Indicator, Vulnerability, Course-of-action, Tool, Attack-pattern, Campaign, Filename, Hash, IP Addresses. |
| Rahman et al. (2024) | Dataset of CTI reports. | Unstructured | 7,052 sentences (94 CTI reports). | – |
| Høst (2022) | NVD | Both | 150,000 CVE records. | CVE ID, CWE ID, description etc. |

of nodes are clustered or separated, as well as their tendency to disassemble or strengthen. One of the community algorithms is the Label Propagation algorithm (LPA), it is a fast algorithm and is used to find communities in the graph. The benefits of using this algorithm are that discovers the communities using only network structure as guidance, and neither needs a pre-defined objective function nor previous information about the communities. LPA propagates the labels over the graph to form communities of nodes depending on their influence.

Husák et al. (2023) posited the provision of network-wide cyber situational awareness particularly its comprehension level by graph-based analytics. They depend on different monitoring and reconnaissance tools to collect information at the right time on a computer network and devices such as network information, vulnerabilities, and CVEs. Then, they used graph-based analytics such as Neo4j Graph Data Platform to store and visualize the collected data. They used graph-based analytics to attain operational CSA in practice and eventually ease the network defenses preparation, preventive actions planning, and accelerate incident responses and network forensics.

Sun et al. (2023) proposed SecTKG, which is an automated architecture for constructing knowledge graphs for open-source security tools. An ontology model was created to

characterize the properties of security tools, their users, and their associations within the open-source community. A knowledge graph based on the security tools' ontology can be applied in practice using some graph algorithms. As a result, they developed a method for measuring the influence of security tools using multiattribute decision-making and linear threshold propagation. In order to find some security tools that are high-quality, well maintained, but less famous, this method examines node attribute information, different types of relationships, and social network analysis.

Tovarňák, Sadlek & Čeleda (2021) suggested an elegant graph-based method for matching vulnerable configurations. This method describes the affected hardware and software logically, taking into account the versions and conditions of the vulnerability. Standards for CPE naming, CPE dictionaries, CPE applicability languages, and CPE name matching are specified in CPE specifications. However, research articles usually only use specifications for CPE naming.

The CPE strings of vulnerabilities and device fingerprints are usually matched using a brute-force approach. However, large-scale implementation requires more efficient algorithms. In a single graph traversal, the authors provide a query to find all matches between vulnerable CVEs and asset configurations from a decomposed CPE string model.

## The refinement of KG

Refinement techniques for KGs encompass rule-based approaches (Dong et al., 2014; Chen, Jia & Xiang, 2020), machine learning-based methods (Lin, Subasic & Yin, 2020; Tiwari, Zhu & Pandey, 2021; Wan et al., 2021), and graph analytics-driven strategies (Aggarwal, 2011; Petermann et al., 2016). Several studies in the literature have employed these techniques to enhance the quality and utility of KGs, ensuring improved accuracy, consistency, and relevance. These techniques have been discussed as below:

Machine learning approaches

Pingle et al. (2019) proposed the RelExt system to improve different cyber threat representation schemes, including cybersecurity knowledge graphs *via* predicting relations between cybersecurity entities determined by cybersecurity NER. While Høst (2022) has used a knowledge graph embedding method (TuckER) to predict missing entities to enhance the initial graph. Where Jia et al. (2018) used a path-sorting algorithm for relationship deduction, where a new relationship can be obtained by using the path connecting two entities as a feature to predict the relationship between the two entities.

Rule-based techniques

Qin & Chow (2019) introduced a reasoning function inspired by association rule mining (ARM), aimed at uncovering hidden rules and inferring relationships between various types of weaknesses. This approach leverages ARM to identify and reason about latent connections within the data.

Beyond machine learning approaches and rule-based techniques, other methods for KG improvement have been explored, such as fusion processes and canonicalization. KG Fusion involves integrating multiple KGs into a unified structure, thereby creating a more comprehensive and accurate representation of the underlying information. In contrast, canonicalization refers to standardizing different mentions of identical entities or

relationships into a single form. This process is essential to reduce redundancy and enhance the overall quality of the KG.

*Sarhan & Spruit (2021)* applied refinement and fusion techniques for KG canonicalization, streamlining the data into a cohesive format. Similarly, *Piplai et al. (2020)* employed fusion methods to integrate knowledge from various AARs that describe the same entity, thereby improving the quality of their CKG.

## Graph representation formats in knowledge graphs

The majority of studies have adopted the LPG as the graph representation format for knowledge graphs (*e.g.*, *Husák et al., 2023*; *Sarhan & Spruit, 2021*; *Sun et al., 2020*; *Sun et al., 2023*; *Wang et al., 2020*). In contrast, some studies have utilized the resource description framework (RDF) format (*e.g.*, *Qin & Chow, 2019*). Although RDF offers advantages such as semantic interoperability and adherence to W3C standards, it suffers from several significant shortcomings. These include challenges with vocabulary standardization, which can lead to inconsistencies when integrating data from multiple sources Additionally, selecting an appropriate syntax format (*e.g.*, Turtle, RDF/XML) and choosing an optimal query language (*e.g.*, SPARQL) can complicate implementation. RDF's inability to natively represent properties on edges further limits its expressiveness in scenarios requiring rich relationship annotations, such as weights or temporal data. Moreover, RDF-based systems often incur higher computational overhead due to the verbosity of RDF triples and the complexity of SPARQL queries.

In contrast, LPG overcomes many of these challenges by enabling properties to be directly associated with edges, providing a more intuitive structure for representing complex relationships and facilitating efficient graph traversal. Given these advantages and the limitations of RDF, this study adopts the LPG format to ensure improved functionality, scalability, and usability of the Knowledge Graph.

## Use cases for cybersecurity knowledge graphs

The concept of knowledge graph gained interest in different fields, particularly cyber security. KGs can deal with large amounts of data created from cyberspace. KGs capture the complexity and heterogeneous nature of information using knowledge representations based on ontologies (*Sikos, 2023*). There are various use cases for cybersecurity knowledge graphs such as vulnerability analysis and intrusion detection.

*Kiesling et al. (2019)* provided a query-based scenario to show how the alerts of network intrusion detection system (NIDS) can be linked to the SEPSES CSKG to obtain a better knowledge of current attacks and possible threats. Also, they provided another use case for the SEPSES KG, which is based on query, to show how their KG can assist security analysts by creating the information organization-specific asset to a chain of well-known vulnerabilities that is frequently updated.

*Chen (2020)* proposed a DDoS attack detection technique based on a domain KG. This technique primarily targeted at DDoS attacks on TCP traffic. The TCP traffic communication process between two hosts is expressed using the KG. In addition to compute the one-way transmission propensity's value and a threshold was determined to

identify if the source host launches a DDoS attack. *Li et al. (2022)* proposed a method called AttacKG for constructing the knowledge graph. This method can efficiently detect network attacks by automatically extracting content from CTI reports. They also provided a new concept called Technique Knowledge Graph, which sum up the causal techniques from attack graphs to describe the complete chain of attack in CTI reports. They provided two use cases in which TKG can be used, TKGs use the collected knowledge to enhance reports, which assist in understanding and reconstructing certain attacks. Furthermore, TKGs with aggregated technique-level intelligence can improve the attack variants' detection.

## MOTIVATION FOR AUTONOMOUS CONSTRUCTION AND ANALYSIS

Currently, there is a variety of knowledge graphs available for cybersecurity. Most of the previous studies used unstructured data such as security news, blogs, and CTI. While a few research had used structured data. However, only *Sun et al. (2023)* and *Qin & Chow (2019)* use heterogeneous data, therefore, there is a lack of research that uses heterogeneous data. Also, most of the existing solutions did not include graph-based analytics, which made them unable to gain deeper insights into the data. In addition, most of the existing KGs may consist of low-quality items such as incorrect relations, and erroneous entities. As they simply fed the data into KGs directly by mapping and did not refine the KGs, thus, their KGs did not have consistency and accuracy.

Consistency in a knowledge graph means maintaining logical coherence in which data should align without contradictions, and relationships must be meaningful according to established rules or ontologies. Ensuring consistency may involve steps like error correction, resolving conflicts, and unifying terms across datasets (*Hur, Janjua & Ahmed, 2021*). Accuracy, on the other hand, reflects how well the KG mirrors real-world entities and relationships, capturing relevant data accurately and filtering out irrelevant or incorrect information. Most existing approaches focus heavily on the construction of KGs but lack sufficient refinement, which can lead to lower overall quality. Specifically, existing solutions often do not adequately address consistency and accuracy, creating a significant gap in KG reliability.

Minimal efforts in refinement may lead to inaccuracies or inconsistencies that undermine the KG's utility. Consistency can be measured through logic-based validation (*e.g.*, identifying conflicting triples or detecting structural anomalies), while accuracy can be evaluated by assessing alignment with ground truth data or expert-curated annotations. While automated refinement processes can help KGs achieve a degree of consistency and accuracy, achieving high accuracy in dynamic fields like cybersecurity often requires domain-specific rules and continuous updates to address the evolving nature of threats (*Hur, Janjua & Ahmed, 2024*).

In this work, we extract the unstructured data from an information extraction system to feed the extracted information into a knowledge graph, where the entities as nodes and the relations as edges associated between the nodes. Information extraction systems can extract such data, and these extractions were used to construct a knowledge graph.

Generally, entity extraction is considered as a difficult task, many textual references that initially look different may indicate the same entity. However, some textual references also incorrectly correspond to different nodes. As knowledge graphs' current knowledge usually consists of data insufficiency, redundant information, inconsistencies, and incomplete data that result in the lack of CKG coverage. Only a few prior studies employed diverse reasoning methods, such as knowledge graph embedding and techniques inspired by association rule mining (*Qin & Chow, 2019*; *Sarhan & Spruit, 2021*). In contrast, our approach refines the KG by incorporating logical rules as a reasoning mechanism. This innovative method effectively identifies hidden links within the KG, significantly enhancing its coverage.

Information obtained from different sources can increase the knowledge in the KG as there is a significant need to create a comprehensive CKG by employing structured and unstructured data sources (*i.e.*, heterogeneous data) in the KG construction. As unstructured data usually consists of valuable information that can be analyzed by using NLP techniques. Most of the previous studies tackled either structured or unstructured data to construct the KG. While we proposed autonomous knowledge graph construction from heterogeneous sources based on the LPGs. Also, we used different graph-based analytics algorithms, which is a powerful feature to analyze and obtain a comprehensive understanding of data in the KGs.

# CYBER GUARD GRAPH APPROACH

The primary goal of this section is to provide a description of our Cyber Guard Graph approach.

## Overview of the cyber guard graph approach

Figure 3 illustrates the workflow of the Cyber Guard Graph approach, encompassing stages from data collection to KG construction and refinement. The process begins with the collection of semi-structured data in the form of NVD documents, which include CVE and CWE entries. For unstructured data, we developed and implemented a robust NLP pipeline. This pipeline includes essential tasks such as text preprocessing (*e.g.*, tokenization and syntactic parsing), named entity recognition and classification, named entity linking and disambiguation, and semantic role labeling. These steps enrich the KG with valuable insights and semantic relationships extracted from the unstructured data.

In handling structured data, we mapped source database items to KG elements by converting data items into nodes, edges, and their attributes. Key-value pairs from CVE JSON files were directly integrated into the KG, preserving the structure and semantics of the original data. During the refinement stage, we applied logical rules to enhance the quality and consistency of the KG. This stage ensures the removal of redundancies and establishes stronger semantic coherence within the graph. The final output of this process is the CKG, a comprehensive and enriched representation of the data. To evaluate the performance and utility of the CKG, we employed various graph analytics algorithms. These algorithms enable the assessment of structural properties, relationships, and the overall effectiveness of the graph in representing cybersecurity knowledge.

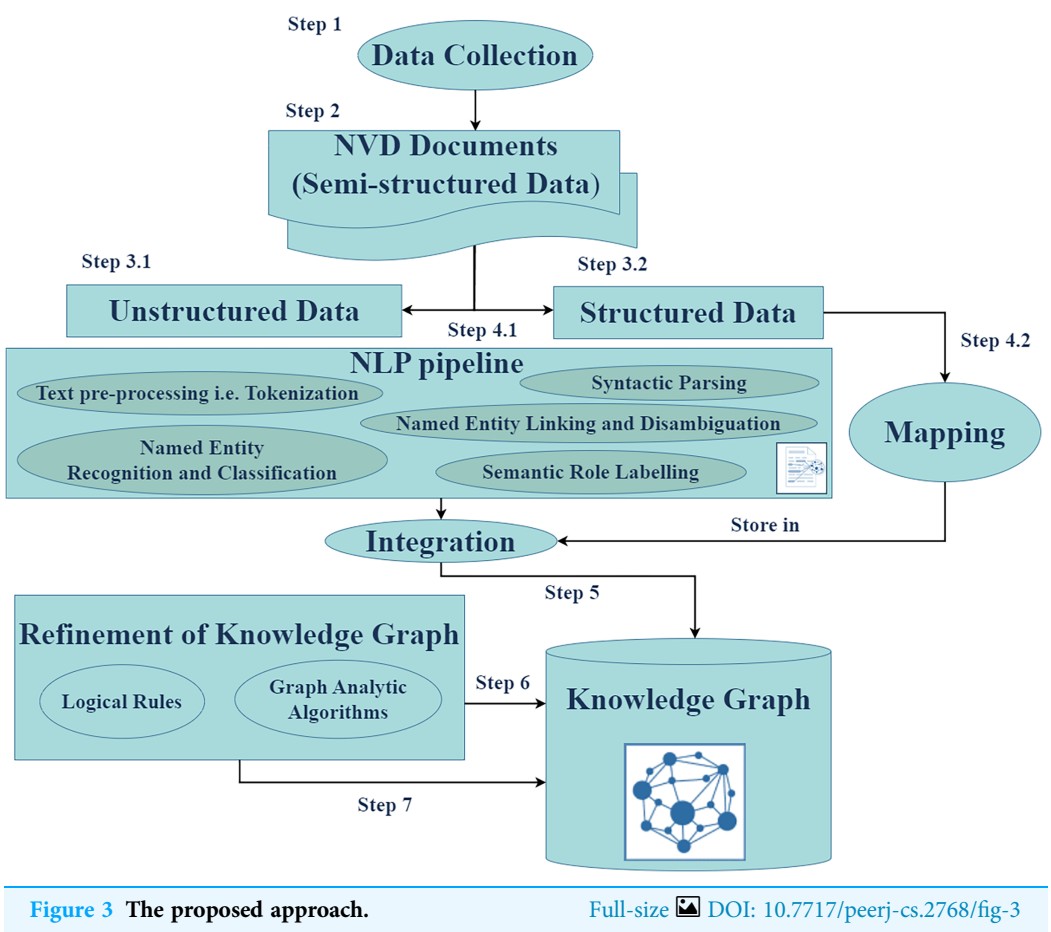

**Figure 3** The proposed approach.

## Data model of cyber guard graph approach

Figure 4 presents a high-level conceptual data model for the Cyber Guard Graph approach, illustrating the relationships and structure of key entities within the knowledge graph. Each CVE entity includes a unique identifier (ID) that specifies the vulnerability and its assigner. A connection is established between CWE nodes and CVE nodes, representing the weaknesses that contribute to specific vulnerabilities, such as cross-site scripting. The ProblemTypes node contains attributes such as lang and value, which correspond to CWE IDs. Each CWE entity includes a CWE ID that is MAPPED_WITH its Extended_Description. To represent related weaknesses within the graph, a HAS_RELATED_WEAKNESS relationship is created, linking CWE nodes and capturing detailed information about their interconnections. The Extended_Description node holds textual descriptions of weaknesses and is connected to Entities *via* a HAS_ENTITY relationship. Additionally, Common Platform Enumerations (CPEs) include attributes such as cveId and version, providing information about the impact of a specific CVE on a product.

The impact entity contains CVSS data, which is critical for assessing the severity of vulnerabilities. A relationship between CVSS nodes and CVE nodes reflects the severity of vulnerabilities based on CVSS scores. This relationship is essential for understanding and

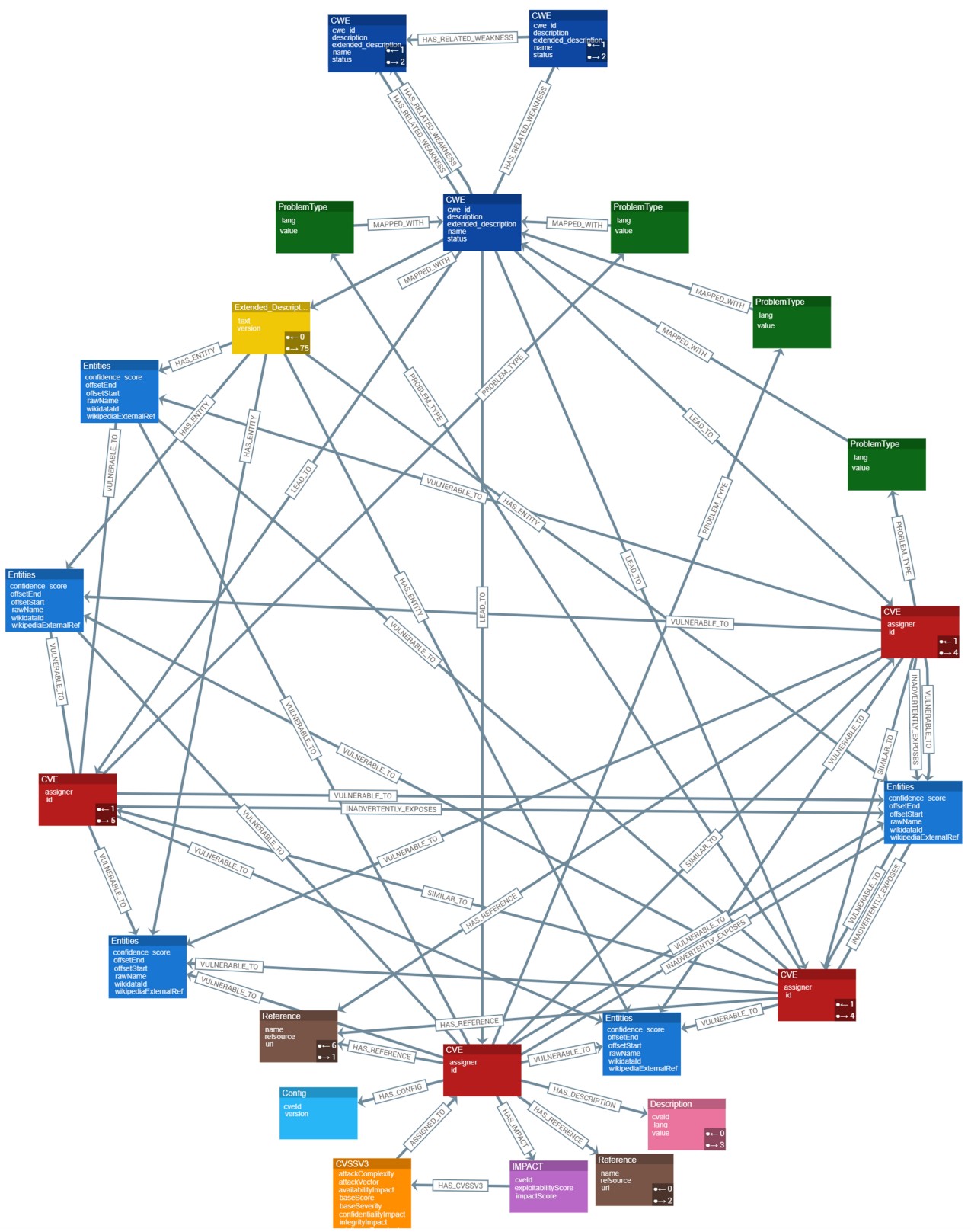

**Figure 4 Data representation and schema design for CKG.**

prioritizing vulnerabilities. To ensure the integrity of the graph, each CVE node is assigned a unique ID property, guaranteeing accurate identification and traceability within the KG.

## Methodology for CKG generation

The methodology presented is a proof-of-concept (PoC) implementation of the Cyber Guard Graph approach, specifically designed to operate within a graph database. This PoC demonstrates how the methodology can be applied to cybersecurity data, illustrating its practical implementation in building a KG. By showcasing this PoC, we highlight the feasibility and effectiveness of the Cyber Guard Graph approach in structuring and analyzing cybersecurity data within a graph-based system. This approach is technology-independent, allowing for its adaptation to any graph database or system capable of storing graph data. Therefore, the proposed solution can be easily transformed and applied to different tool- specific technologies, offering flexibility and broad applicability. The framework we followed is illustrated in Fig. 5. Our methodology accommodates a wide range of cybersecurity data sources, whether structured, unstructured, or semi-structured, enabling the construction of a comprehensive vulnerability KG from these diverse data types. The data collected from NVD documents is semi-structured, containing both structured and unstructured components.

The structured data sources, including CVE and CWE, are directly mapped into the KG. We also parsed CPE values from CVE data to integrate additional information. However, the data also contains significant unstructured components, such as the vulnerability descriptions in CVE records and extended descriptions in CWE records. For example, the CVE-ID feature represents a unique identifier assigned by CVE, along with the current description of the vulnerability and post-analysis information. The records also include references to external advisories, solution methods, and related tools, providing crucial context for vulnerability management. In addition, severity features are associated with CVSS scores and vulnerability vectors, helping to assess the impact of vulnerabilities. Other important features include weakness enumeration categories (such as CWE-ID, CWE name, and source), as well as configuration details of affected software and versions. The CPE values parsed from CVE data include details such as vendor, product name, and version number. The CWE entities themselves contain several attributes, including the CWE ID, name, abstraction level, status, description, extended description, related weaknesses, and other key features.

To extract meaningful information from the unstructured text, we applied NLP techniques such as tokenization, syntactic parsing, named entity recognition, and SRL. These processes involve identifying entities, relationships, actions, and arguments within the text. Entity linking and disambiguation are also critical steps, where entities are linked based on their contextual relationships, as identified through SRL. SRL further aids in tasks such as extracting relations and events, thereby contributing to the structured representation of data within the KG.

Once the unstructured text has been processed and structured, the extracted information is mapped to the KG schema, enriching the graph with relevant entities,

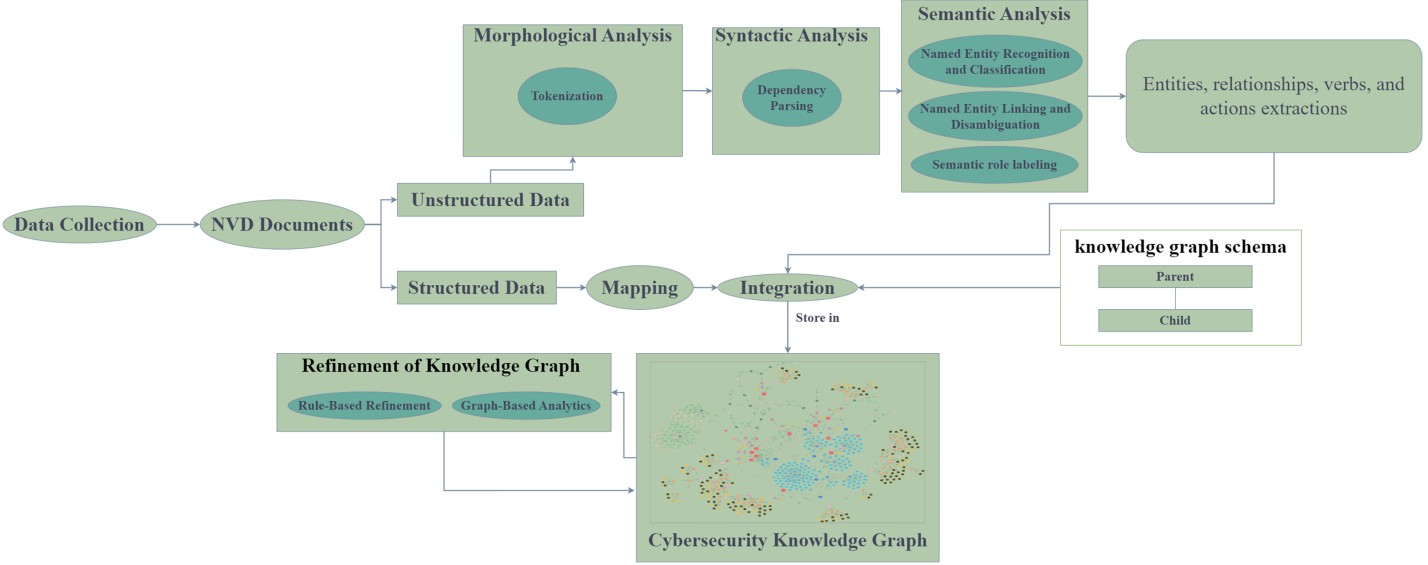

**Figure 5 The proposed framework.**

relationships, and attributes. The KG is then populated, integrating data from both structured and unstructured sources.

In the refinement stage, we enhanced the coverage and quality of the KG through logical reasoning methods. Specifically, we applied deductive rules to infer new knowledge and validate the graph's consistency. Finally, the KG was analyzed using graph analytics algorithms, including centrality and community detection, to evaluate the structure and effectiveness of the knowledge graph in representing cybersecurity data. These analytics provided insights into the importance of nodes and the identification of clusters within the data, further enhancing the understanding of vulnerability relationships.

For analyzing the unstructured content in the datasets, we employed a three-phase process of natural language understanding: morphological analysis, syntactic analysis, and semantic analysis. Morphological analysis involves tokenizing the text, removing punctuation, and analyzing individual words. Tokenization is a crucial step in NLP, as it splits paragraphs and sentences into smaller units that are easier to interpret. The output of the morphological analysis feeds into the syntactic analysis, which structures the sentence by categorizing words and forming syntactic relationships. The semantic analysis phase assigns meaning to the syntactic structures, ensuring that the processed text is mapped to the relevant tasks and entities.

Through this methodology, we effectively transformed raw cybersecurity data into a structured, semantically rich KG, ready for further analysis and decision-making.

Data collection from NVD documents, CVE, CWE, AND other sources

In this article to provide a comprehensive KG, we used different data sources containing structured and unstructured data. Usually, structured data contains numbers and text that are stored in a relational database. While unstructured data contains text or video and is stored in its native format. Besides structured and unstructured data, there is a third category that combines both. A semi-structured data has some defining characteristics but

does not conform to a structure as rigid as is expected with a relational database. For data, it is easy to organize structured data and follow a rigid format; unstructured is complex and often qualitative information that cannot be reduced to or organized in a relational database, and semi-structured data contains elements of both. We used vulnerabilities-related datasets including CWE and CVE provided by NVD documents, these datasets are available in different file formats such as JSON. CVE dataset is a dataset that contains vulnerabilities entries. CVE contains exposed vulnerabilities, and the list of vulnerabilities is preserved by MITRE. CVE is considered a standard to identify and name certain vulnerabilities. While the CWE dataset is a dataset that contains a list of weaknesses and vulnerabilities. CWE basically provides information on the causes of vulnerabilities where every CWE represents one type of vulnerability, and it is considered a standard to classify and describe the weaknesses' types which lead to vulnerabilities. The construction of the KG was done by using Neo4j, which is a high-performance graph database management system used to store nodes and relationships and to visualize and query the KG.

Processing semi-structured data from databases

We obtained the CVE dataset from NVD documents. We first selected NVD because it is a well-known database that provides datasets related to the vulnerability field. This database links with other data sources such as CWE and CPE. As shown in Fig. 6, every entry in the dataset contains CVE, problem type, references, description, configurations, and impact. CVE contains an ID that represents a certain vulnerability and ASSIGNER, problem type contains a value that represents the CWE id. The configuration contains CPE that presents information about the products impacted by a particular CVE. Whereas impact consists of CVSS information, CVSS is used to evaluate vulnerabilities' severity.

Processing structured data

The object CVE nodes contain a unique identifier property which is cve id. So, we need to create a constraint CVE called cve. Because it consists of a lot of information that is required for our graph.

We noticed that the problemtype nodes in KG only provide CWE ids. Thus, to enrich the KG, we needed further information about CWE since CWE serves as a dictionary of software vulnerabilities. CWE contains an ID that represents a specific weakness a name that represents the weakness' name. While description provides a brief description of weakness. Whereas extended description includes a detailed description of weakness. Moreover, a CVE entry contains CPE information. As shown in Fig. 7, the CPE offers a standardized string that is in the form of uniform resource identifiers to identify which products and versions are vulnerable.

The first part of URI identifies that it is a CPE and its version. Then, a part refers to h, a, or o for hardware, application, or operating system respectively. The consequent fields are employed to identify the component by determining the vendor, the product name, the version number, and so on. Since the CPE consists of valuable information, it is an important task to parse the CPE values from CVE data to enrich the KG.

Processing unstructured data

As we mentioned above, the semi-structured data sources including CVE and CWE were imported directly into the KG. Nevertheless, they consist of unstructured data (*i.e.,*
```
"cve" : {
    "CVE_data_meta" : {
        "ID" : "CVE-2023-0012",
        "ASSIGNER" : "cna@sap.com"
    },
    "problemtype" : {
        "problemtype_data" : [ {
    "value" : "CWE-284"
        } }},
    "description" : {
    "description_data" : [ {
        "value" : "In SAP Host Agent (Windows) - versions 7.21, 7.22, an attacker who gains local membership to
SAP_LocalAdmin could be able to replace executables with a malicious file that will be started under a privileged
account. Note that by default all user members of SAP_LocaAdmin are denied the ability to logon locally by security
policy so that this can only occur if the system has already been compromised."
        } ]}
    },
    "configurations" : {
        "cpe23Uri" : "cpe:2.3:a:sap:host_agent:7.21:*:*:*:*:*:*:*",
        }, {
        "cpe23Uri" : "cpe:2.3:a:sap:host_agent:7.22:*:*:*:*:*:*:*",
    } }, {
        "cpe23Uri" : "cpe:2.3:o:microsoft:windows:-:*:*:*:*:*:*:*",
    },
    "impact" : {
        "baseMetricV3" : {
        "cvssV3" : {"version" : "3.1","vectorString" :
"CVSS:3.1/AV:L/AC:L/PR:H/UI:N/S:U/C:H/I:H/A:H","attackVector" : "LOCAL","attackComplexity" :
"LOW","privilegesRequired" : "HIGH","userInteraction" : "NONE","scope" :
"UNCHANGED","confidentialityImpact" : "HIGH","integrityImpact" : "HIGH","availabilityImpact" :
"HIGH","baseScore" : 6.7,"baseSeverity" : "MEDIUM"
    }
```

**Figure 6 CVE data entry.**

```
cpe:2.3:part:vendor:product:version:update:edition:
language:sw_edition:target_sw:target_hw:other
```

**Figure 7 Uniform resource identifiers.**

long sentences). The unstructured data includes the vulnerability description in CVE data and the extended description in CWE data. It is important to analyze the vulnerability description as it contains important information on the vulnerability including how attackers may exploit this vulnerability, how this vulnerability affects users, operating systems, or applications, and possible recommendations. Additionally, the extended description also contains important detailed information on the weaknesses and how they occurred. For unstructured data, we used different NLP techniques to analyze these unstructured contents to extract data from the text.

Open information extraction techniques

Open information extraction (Open IE) are popular techniques, which are used to convert unstructured text to structured text. These techniques extract a big set of relational triples from unstructured text without human interference nor require domain expertise. Then, the extracted information is used to construct a knowledge graph. OIE techniques are categorized into learning-based, rule-based and inter-proposition-based systems. Taking "John managed to open door" as an example, an Open IE extractor should produce the tuple (John; managed to open; the door) but it is not necessary to produce the

Alharbi et al. (2025), *PeerJ Comput. Sci.*, DOI 10.7717/peerj-cs.2768 19/46

extraction (John; opened; the door). In response to this initial task presentation, Open IE has received substantial and consistent attention. There were many automatic extractors created (*Muhammad et al., 2020*). We basically provided the unstructured text to the Open IE model. The outputs of this model are a set of extractions. Each of these extractions is tagged such that contains one predicate (*i.e.*, verb) and a number of arguments. By using this model, the objective is to extract entities based on open information extraction to get the head, which is the most important word of the sentence, so that we can normalize or create a node for the head to determine whether it is an entity or verb. We have created a set of patterns which operates on the dependency parse information of the text.

Graph patterns are considered constraints on subgraphs, and they are the most important component of Cypher queries. Graph patterns depict the data to be retrieved as nodes and edges. Inside a MATCH clause, graph patterns are used to define the data we are searching for. For instance, the MATCH clause uses a built-in index to retrieve all nodes labeled as vulnerability. We will discuss patterns in more detail in the section Refinement of CKG. So, we are following of existing approach of identifying the headword based on the dependency parse information. Since there are existing systems, dependency parsing is used to extract the headwords.

Tokenization

By tokenizing, we mean separating information into tokens. Taking the input text and breaking it up into its fundamental units called tokens, they usually consist of words, numbers, and symbols, divided by white space. Many sophisticated algorithms use tokens as inputs in applications involving linguistic analysis, so tokenization is an essential step input rather than raw text. Tokenizers have to be of high quality in order to avoid problems in other components of the pipeline. A token can be classified according to its capitalization degree, such as a number, a mark, a punctuation mark, a word, *etc.* (*Maulud et al., 2021*). Figure 8 illustrates an example of tokenization. Another example from our data is that "An attacker who gains local membership to words are SAP_LocalAdmin" was identified as the headword and classified as a noun by the spaCy model. All other dependent on the head word "an attacker". There is an "who" tagged with PRON, a "gains" tagged with VERB, a "local membership" tagged with NOUN, an "to" tagged with ADP, and a "SAP_LocalAdmin" tagged with PROPN.

Syntactic parsing

Dependency grammar is a part of syntactic text analysis. Dependency grammar identifies the root word, which means the word has no dependency on other words in a given sentence, and the method also identifies important relationships (*e.g.*, the nominal subject), these grammar relationships can be used for entity identification. The other words are given a label to represent the relationship they have with either the root word or another word that is adjusting the root. Consider, for instance, the sentence "John is playing a game". As a result of parsing it, the components will be listed as "John", "is", "playing", and "game". Natural language processing parses a natural language sentence using the same concept. Normally, parsing a sentence in natural language involves analyzing its grammatical elements, identifying its parts of speech, *etc.* Therefore, "John",

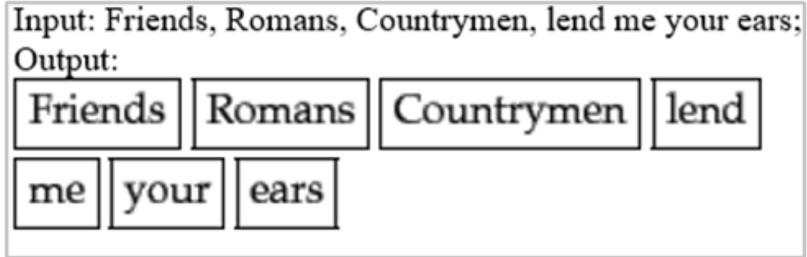

**Figure 8 Example of tokenization.** 

"is", "playing", "game" are tokens that refer to the above sentence. In every natural language, the sentences are formed according to their own grammar rules. This is known as "syntactic relations" (*McIntyre, 2021*). Our data emphasizes the syntactic structure of the sentence. For instance, "An attacker who gains local membership to words are SAP_LocalAdmin". As a result of parsing it, the components will be listed as "An attacker", "who", "gains", "local membership", "to", "SAP_LocalAdmin". The tokens that refer to the above sentence are "An attacker", "who", "gains", "local_membership","SAP_LocalAdmin".

Named entity recognition and classification

Named entity recognition and classification (NERC) involves identifying proper nouns in text and classifying them as different kinds of named entities (NE), such as individuals, organizations, locations, *etc*. For example, Sundar Pichai, CEO of Google Inc., walks through the streets of California. Based on the preceding sentence, we can categorize things into three different kinds ("person": "Sundar Pichai"), ("organization": "Google Inc."), and ("state": "California"). Another example from our data also shows that "The software constructs all or part of a code segment using externally-influenced input from an upstream component, but it does not neutralize or incorrectly neutralizes special elements that could modify the syntax or behavior of the intended code segment." As a result of the preceding sentence, we can categorize things in three categories: There are different types ("weakness_abstraction": "Base"), ("name": "Code Injection"), and ("status": "Draft"). It is an important subtask in several language engineering applications, especially in the field of information retrieval and extraction. NERC systems have to be validated on enormous corpora, so they are capable of recognizing and categorizing NEs correctly (*Petasis et al., 2001*).

Named entity linking and disambiguation

Named entity recognition and disambiguation (NERD) is also known as entity linking (EL), which used to identify named entities in plain text and determine their meanings. The NER task starts by indicating the named entities' text span. Then, named entity disambiguation (NED) task comes to link the text span of each named entity to the correct entry in a knowledge base. NERD technique is an fundamental in various NLP applications, because this technique assists in extracting and understanding the information from plain text. The goal of NER is first to identify the text spans of entities.

NED then links the text spans of each named entity to the correct entry in a knowledge base (*Lin, 2021*). For example, the statement "a malicious file" that will be started under a privileged account. Entities were extracted from a vulnerability description by the model. "a malicious file" labeled with ARG1, which stands for Argument 1. "that" tagged with R-ARG1, that refers to Reference Argument 1, "will" tagged with ARGM-MOD, that refers to Modifier Modal, "started" tagged with V which stand for to Verb (predicate), and "under a privileged account" tagged with ARGM-LOC, that refers to Modifier-Location.

Semantic role labeling

A shallow semantic parsing task called semantic role labeling (SRL) identifies all constituents that fill a semantic role for each predicate in a sentence and determines their roles (Agent, Patient, Instrument, *etc.*) and adjuncts (Locative, Temporal, Manner, *etc.*). SRL is also capable of extracting structured information from unstructured text. In addition to providing deeper understanding of sentence semantics, SRL facilitates downstream tasks such as answering questions, extracting information, and translating texts. An example of a sentence would be "John ate the apple," where "John" is the subject, "ate" is the verb, and "the apple" is the object (*Punyakanok, Roth & Yih, 2008*). Another example is "a malicious file that will be started under a privileged account", where "file" is the subject, "started" is the verb, and "the account" is the object.

## Refinement of CKG

In the field of KG, there are many reasoning methods proposed to enhance the coverage of KG. Generally, knowledge reasoning means a way to draw conclusions from known facts to infer new knowledge, one of the reasoning methods is logical rules (*i.e.*, deduction rules).

Integration of deduction rules for KG refinement

The reasoning methods are used to enrich the KG, particularly, logical rules used to improve KG by deducing new relations between two unrelated entities with each other. We employed logical rules to enhance the KG coverage. The graph pattern describes data using a syntax that is like how the nodes and relationships of a property graph are drawn on a whiteboard, where the nodes are represented as circles and the relationships as arrows. So basically, we have the conditions, and those conditions imply a new fact or new a state, maybe there are connections between two nodes. So, it is having (precedent→consequent) rule head→rule bod This is how we create the rules. For example, a new relationship between two nodes that were not initially present but based on the current state of data, we observe that these two nodes should be connected. So, the reason why they should be connected is that the patterns indicate or give us a clue that these two nodes should be present in our graph. Therefore, patterns are necessary for us to discover new knowledge. It will help us gain a deeper understanding of data if we discover hidden patterns. Using the knowledge graph, we can uncover fraudulent activities or find alternative actions that can prevent risks by discovering patterns between nodes. The components of these patterns are nodes and relationships. By combining these nodes and relationships, simple or complex patterns can be expressed. In a simple pattern, we create two nodes connected by a single relationship. For example, the weaknesses [:LEAD_TO] vulnerabilities. Whereas in complex patterns, using we create several relationships with many nodes. For instance, the

vulnerabilities that are [:VULNERABLE_TO] attacks and [:SIMILAR_TO] the vulnerabilities that are exploited by attackers to perform the same attack. Those patterns convey complex concepts and support different use cases. We obtained the deduction rules from simple and complex patterns. The following Table 2 illustrates the deduction rules and their descriptions.

Graph analytic algorithms

A graph algorithm is a subset of a graph analytics tool. The concept of graph analytics is using graph-based methods to analyze data connected to a graph. A graph algorithm is one of the most powerful approaches to analyzing connected data since it uses mathematical calculations that are specifically designed to handle relationships. For instance, social networks like Facebook and LinkedIn are all about connections. Social network features depend heavily on graph analytics, such as recommendations of people who are interested. The purpose of using graph analytics is to look at the structure of our data instead of just the data points. This allows us to uncover patterns and predictive information, and it allows us to infer meaning from connections. So, it is about finding patterns that are important to us. Through graph analytics, we can make sense of our connected data by understanding its structure.

## IMPLEMENTATION

We have successfully implemented the rule-based reasoning framework, which provides a robust foundation for deriving insights from the knowledge graph. The implementation section includes a comprehensive specification of the required programming languages, libraries, and tools essential for its development and integration.

### Technology stack for building a proposed cybersecurity knowledge graph

This research utilizes a comprehensive technology stack to construct a cybersecurity KG. In the data collection phase, we first obtained CVE data from the NVD website in JSON format. Detailed information about CWEs was retrieved using the cwe2 library, which was also downloaded as a JSON file (*Hany et al., 2020*). Additionally, CPE values were parsed from the CVE data using the cpe-parser library (version 0.0.2), the latest available CPE parser in Python (*Shukurov, 2022*). The parsed CPE values were stored in JSON format and subsequently imported into the KG. To store the data, we employed Neo4j and the cypher query language, representing the information as entities and relationships within the graph. CVE data, CPE values, and CWE data were then integrated to create a comprehensive KG.

For unstructured data, we applied NLP techniques, including OIE and entity-fishing. OIE was used to convert unstructured content into structured information, while entity-fishing involved named entity recognition, classification, disambiguation, and linking, enabling the extraction of concepts, individuals, and their relationships. We utilized Neo4j Desktop (Community Edition, version 1.5.7) to retrieve data from the graph through Cypher queries. These queries used pattern- based rules to enhance the KG by matching co-referent entities and determining their similarities. The implementation required

**Table 2 Deduced rules and their descriptions.**

| Deduced rule | Description |
|---|---|
| IF PROBLEM_TYPE (cve, problemType) && MAPPED_WITH (problemType, cwe)<br><br>THEN LEAD_TO (cwe, cve) | This inferred relation is obtained by using multiple relationships to connect the CWE nodes with CVE nodes, this relation reflects the weaknesses (*i.e.*, errors) that lead to vulnerabilities. |
| IF PROBLEM_TYPE (cve, problemType) && MAPPED_WITH (problemType, cwe) && MAPPED_WITH (cwe, cwedata) && HAS_ENTITY (cwedata, Entities) && VULNERABLE_TO (Entities, cve) && SIMILAR_TO (cve1, cve2)<br><br>THEN SIMILAR_TO (cve2, cve3) | This deduced relation is obtained by using multiple relationships to connect the attack nodes with CVE nodes, this relation refers to the vulnerabilities that are vulnerable to attacks. We have another relation links between CVE nodes, this relation refers to similar CVE nodes that are exploited by attackers to perform the same attack. |
| IF PROBLEM_TYPE (cve, problemType) && MAPPED_WITH (problemType, cwe) && MAPPED_WITH (cwe, cwedata) && HAS_ENTITY (Entities, cve) && INADVERTENTLY_EXPOSES (Entities, cve) && SIMILAR_TO (cve1, cve2)<br><br>THEN SIMILAR_TO (cve2, cve3) | This inferred relation is acquired by using multiple relationships to connect nodes (*i.e.*, sensitive data/information) with CVE nodes, this relation refers to the vulnerabilities that accidentally expose sensitive data/information. |
| IF HAS_IMPACT (cve, impact) && HAS_CVSSV3 (impact, cvssv3)<br><br>THEN ASSIGNED_TO (cvssv3, cve) | This deduced relation is obtained by using two relationships to connect the CVSS nodes with CVE nodes. This relation indicates the severity of vulnerabilities, which these vulnerabilities are assigned by CVSS scores. |

hardware with 32 GB memory, an Intel Core i7 processor, and Windows 10 OS to ensure optimal performance and minimal runtime. In the implementation process, we first wrote a Python script to transform the NVD document into a Python object that contained both structured and unstructured data, particularly useful unstructured descriptions. Next, structured data, including entities, relationships, verbs, and actions, were mapped for input into the graph. Tokenization was carried out using the spaCy library to break down text into smaller components, such as words, numbers, and symbols, making the data easier to analyze and interpret. Specific nodes were created for each token, storing relevant metadata like category, start index, and end index. Syntactic parsing was also performed with the spaCy library, which analyzed the grammar and word sequence in sentences to determine their relationships. Named entity recognition and classification were used to identify proper nouns in the text, which were then classified. Entity-fishing further assisted in linking and disambiguating these named entities. To deepen the analysis, semantic role labeling was performed using the AllenNLP library, which extracted structured information from unstructured text. AllenNLP's flexible API enabled intelligent batching, high-level abstractions for text operations, and provided a framework for rigorous experimentation.

Finally, custom modules for rule-based refinement and graph-based analytics were developed to improve the quality and functionality of the cybersecurity knowledge graph. Figure 9 illustrates a key component of the technology stack, showing how unstructured data was processed using NLP techniques to extract valuable information.

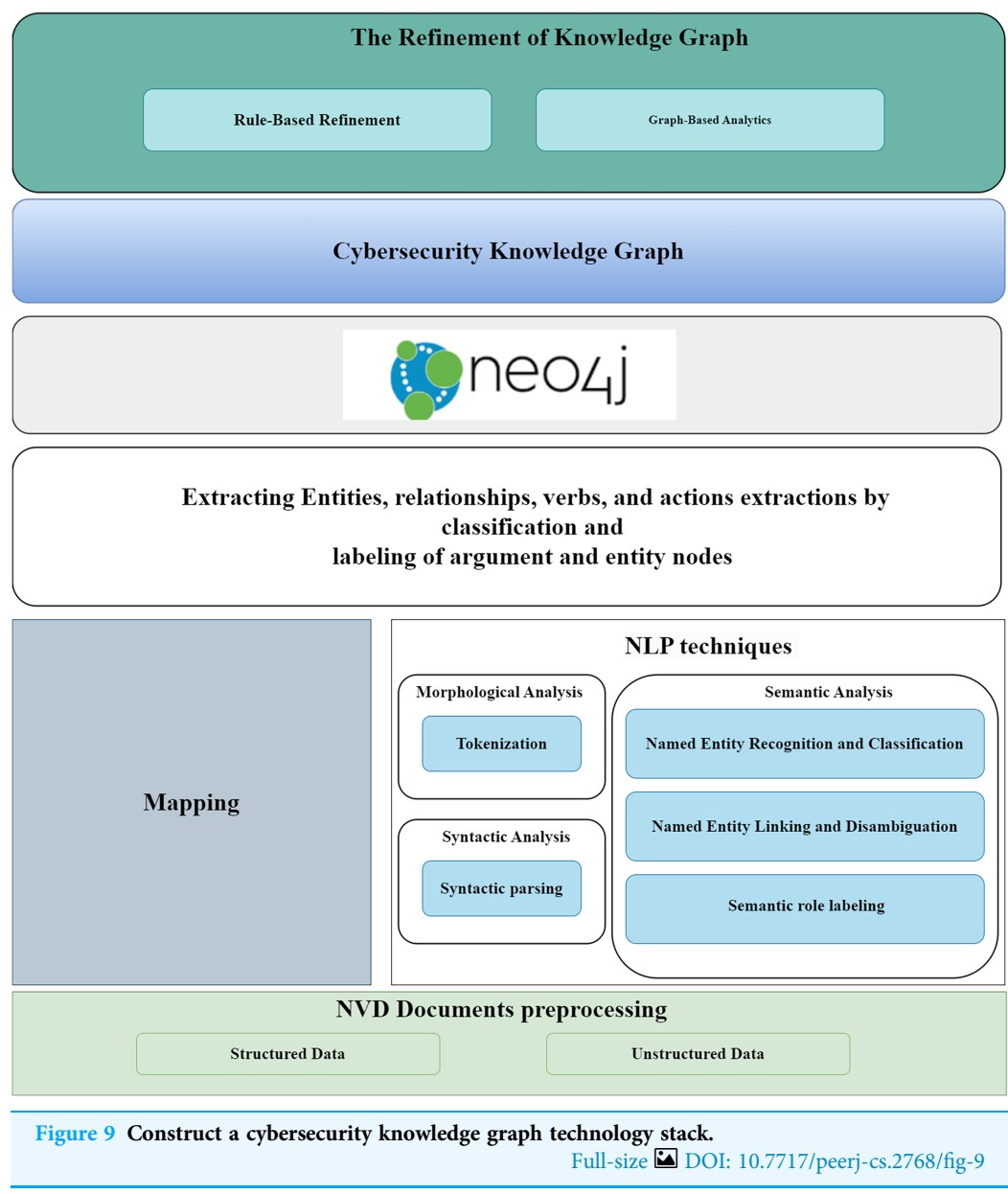

**Figure 9 Construct a cybersecurity knowledge graph technology stack.**

## Data collection and integration process for cybersecurity knowledge graph

The CVE serves as a unique identifier for specific vulnerabilities, enabling security professionals and researchers to pinpoint and discuss these vulnerabilities with precision. In contrast, the CWE identifies the underlying causes, or "weaknesses," that contribute to vulnerabilities. A "weakness" refers to a condition in software, firmware, hardware, or service components that can lead to the introduction of vulnerabilities. CWE is essentially a catalog of common programming and design errors, and a thorough understanding of these weaknesses allows security analysts and developers to address issues at their root, preventing the occurrence of vulnerabilities. Hackers can exploit CVEs to execute code remotely through a particular vulnerability in an operating system. CVE entries focus on

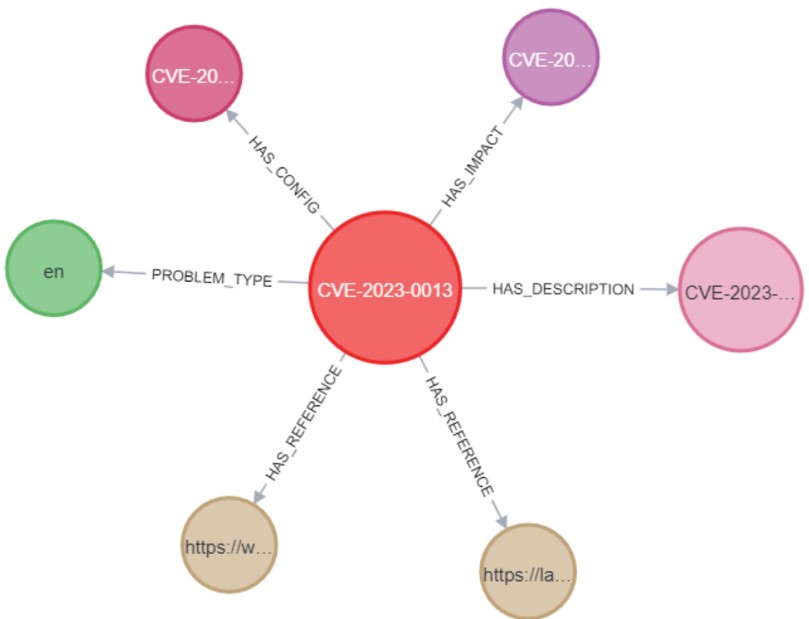

**Figure 10  Example of a CVE node in the KG.**     

detailing a specific product's vulnerability, whereas CWE provides a broader description of vulnerabilities independent of the product involved. As part of our manual data collection efforts, we analyzed and collected CVE and CWE data, which were sourced from vulnerability-related datasets provided by the NVD. These datasets were available in various file formats, including JSON, and were imported into the Neo4j graph database following the procedure outlined in the Cyber Guard Graph approach section.

In the Neo4j database, as shown in Fig. 10, the CVE node was labeled with its unique identifier (*e.g.*, CVE-2023-0013). This CVE node was then connected to other nodes using various relationships. These include the PROBLEM_TYPE relationship, which connects the CVE to its associated problem type; the HAS_REFERENCE relationship, linking the CVE to relevant references; the HAS_DESCRIPTION relationship, which represents the link between the CVE and its description; the HAS_IMPACT relationship, connecting the CVE to its impact; and the HAS_CONFIG relationship, linking the CVE to its configuration details. In Section C, we discussed the implementation of constraints for CVEs, where we assigned unique IDs to ensure that the ID property values for CVE nodes remained unique. This was accomplished by creating a unique CVE property constraint, as shown in Fig. 11. We applied the same type of constraint to all nodes in the graph. Afterward, we created relationships between the nodes using Cypher queries. This process was replicated when importing CWE data into the knowledge graph, ensuring that each relationship between the CVE and CWE nodes was properly mapped.

Through the analysis of the problem type, we identified that some CWE entries contained CWE IDs, which are used to represent the weaknesses associated with vulnerabilities. To enrich the KG, we Gathered more information about the relevant CWEs. Each CWE entry contains a unique ID that identifies a specific weakness, as well as

```
explore_reviews_csv_query = """
CREATE CONSTRAINT CVE IF NOT EXISTS FOR (cve:CVE) REQUIRE cve.id IS UNIQUE
"""
```

**Figure 11  The unique node property constraint.**

```
explore_reviews_csv_query = """
MERGE (problemType:ProblemType {value:"CWE-79"})
MERGE (cwe:CWE {cwe_id:"79"})
MERGE (problemType)-[:MAPPED_WITH]->(cwe)

"""
```

**Figure 12  Mapping between CVE and CWE.**

a name that describes the weakness. We then created Cypher queries to establish the MAPPED_WITH relationship, linking CWE nodes to their corresponding CVE nodes, as illustrated in Fig. 12. This relationship helped associate CVE data with the relevant CWE data. Additionally, we created a HAS_RELATED_WEAKNESS relationship to connect CWE nodes, storing information about related weaknesses in the relationship properties. As shown in Fig. 13, this information includes properties such as CWEFID, NATURE, ORDINAL, and VIEWFID. The NATURE property is particularly useful for identifying the child CWEs of a given CWE, helping to establish the hierarchy of weaknesses. This allowed us to analyz and link similarities between all CWE entries.

After parsing the Common Platform Enumeration (CPE) data using Cypher queries, we created a MAPPED_WITH relationship to connect the parsed CPE data with the CPE match nodes from the CVE data, as demonstrated in Figs. 14 and 15. For processing unstructured text in CVE descriptions, we used OIE, specifically employing the deep bilateral long short-term memory (BiLSTM) sequence prediction model proposed by *Stanovsky et al. (2018)*. This model was implemented using the AllenNLP API (*Gardner et al., 2018*). AllenNLP is an open-source library designed to solve various NLP tasks and offers a variety of pre-trained models that can be utilized for tasks such as semantic role labeling (SRL). In Fig. 16, we show the output of the OIE process. For example, the sentence "a malicious file that will be started under a privileged account" was processed by the model, extracting the entities and labeling them with specific tags such as ARG1 (Argument 1), R-ARG1 (Reference Argument 1), ARGM-MOD (Modifier-Modal), V (Verb), and ARGM-LOC (Modifier- Location). These labeled entities were then used to form relationships between nodes in the graph.

After extracting entities using OIE, we noticed that some arguments consisted of long sentences or text spans. To address this, we used the spaCy dependency visualizer to further analyze the unstructured content, particularly focusing on visualizing the dependencies and named entities within the text. This tool allowed us to determine the head word and its dependent words in each sentence, providing further context for entity extraction and classification. The spaCy parser also enabled us to examine the relationships between verbs in the sentence, which helped refine the analysis of the extracted entities. For

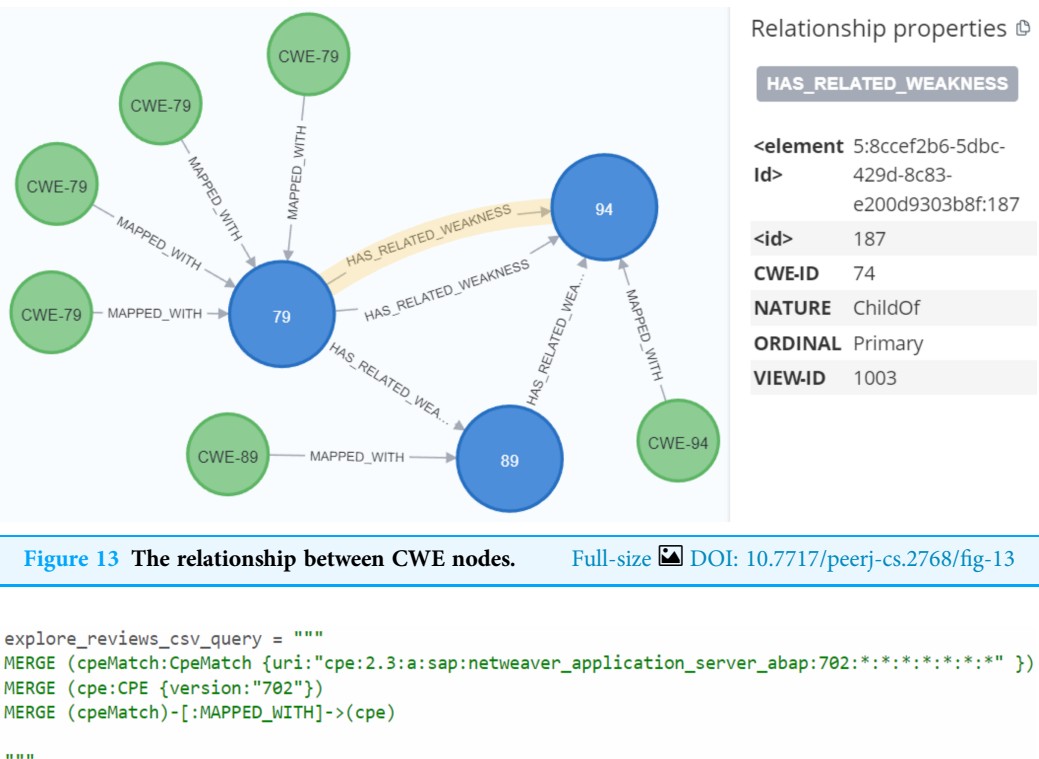

**Figure 13** The relationship between CWE nodes. 

```
explore_reviews_csv_query = """
MERGE (cpeMatch:CpeMatch {uri:"cpe:2.3:a:sap:netweaver_application_server_abap:702:*:*:*:*:*:*:*" })
MERGE (cpe:CPE {version:"702"})
MERGE (cpeMatch)-[:MAPPED_WITH]->(cpe)

"""
```

**Figure 14** Mapping between cpeMatch and cpe. 

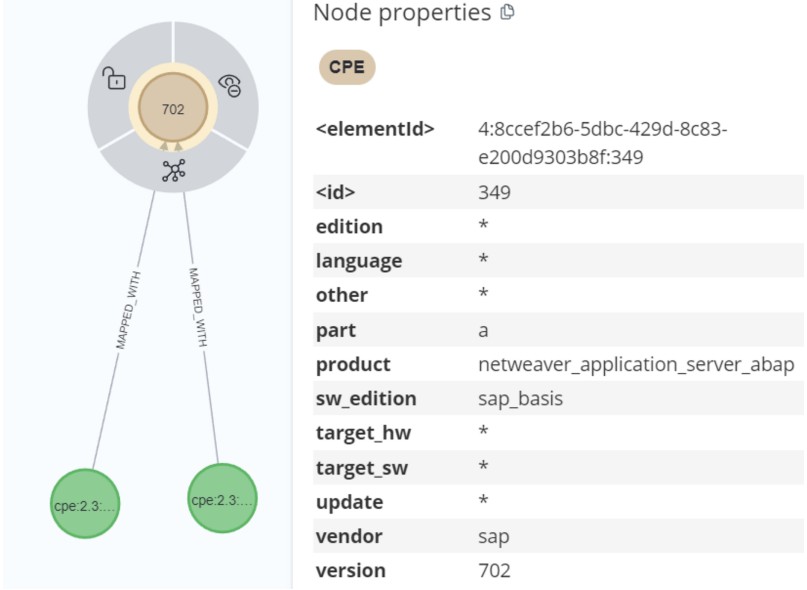

**Figure 15** Mapping between parsed CPE and cpeMatch.

instance, in Fig. 17, we show the output of the spaCy visualizer, which identified "an attacker" as the head word (labeled as NOUN) in the sentence "an attacker who gains local membership to SAP_LocalAdmin." All other words in the sentence were marked as

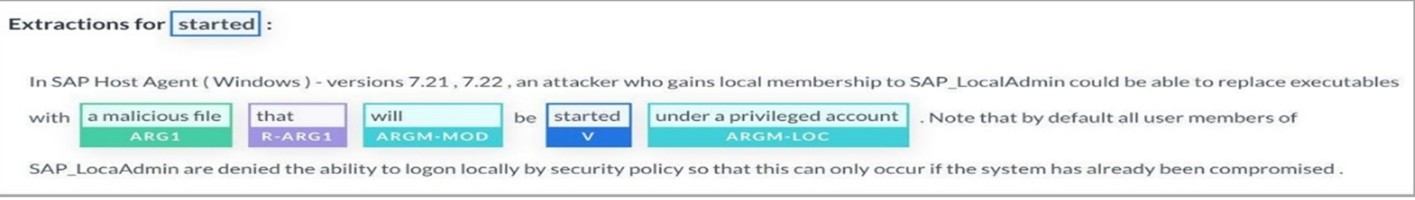

**Figure 16 Entities extraction using open IE.**

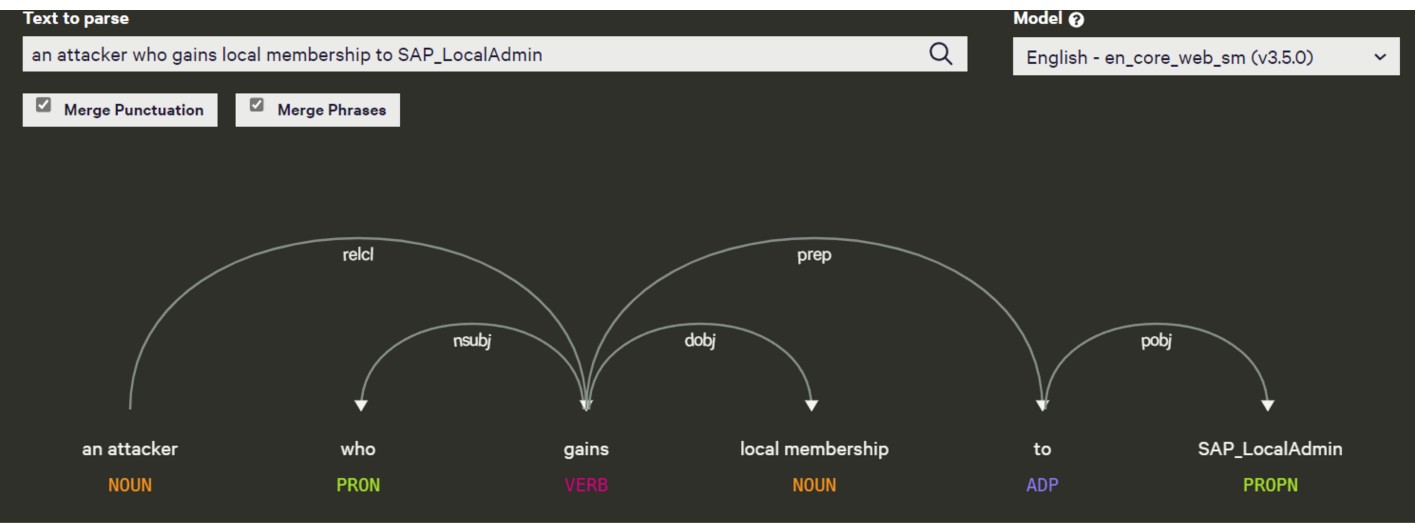

**Figure 17 Dependency parsed sentence using spaCy dependency visualizer.**

dependent on the head word, including "who" (PRON), "gains" (VERB), "local membership" (NOUN), "to" (ADP), and "SAP_LocalAdmin" (PROPN). After parsing the sentence, we created Cypher queries to link the head node to all the dependent arguments using the HAS_ARGUMENT relationship, followed by creating MAPPED_WITH relationships between the vulnerability descriptions and their associated frames (verbs). For unstructured texts in extended descriptions, we utilized entity-fishing, a technique involving named entity recognition, classification, disambiguation, and linking. This tool supports multiple languages and automates the recognition and disambiguation tasks using the Wikidata knowledge base (*Lopez, 2016*). As shown in Fig. 18, we input extended descriptions into the entity-fishing tool, which returned a list of identified and disambiguated entities in JSON format. These entities were then imported into the KG and linked to the corresponding CWE entries using the MAPPED_WITH relationship, as illustrated in Figs. 19 and 20.

## Deduction rule implementation

As illustrated in Figs. 21 and 22, we have provided two examples of the logical rules implemented in our approach. In alignment with the structure shown in Fig. 19, we establish an edge (*i.e.*, relationship) that links weaknesses (CWE) to vulnerabilities (CVE). The edges depicted in Fig. 21 represent vulnerabilities that could inadvertently expose sensitive information. Additionally, another edge is created between CVE nodes to

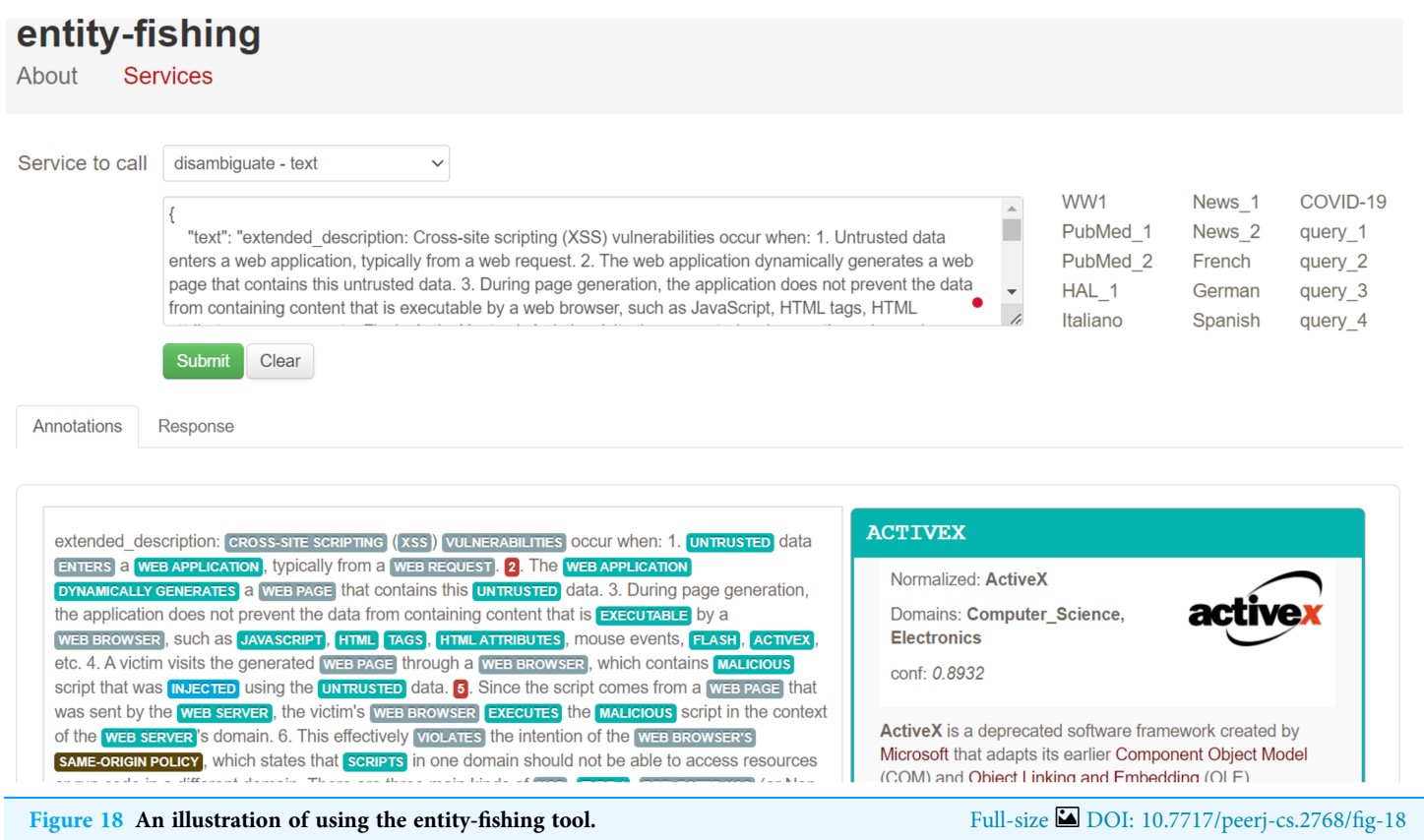

**Figure 18   An illustration of using the entity-fishing tool.**               

```
explore_reviews_csv_query = """
MERGE (cwe:CWE {cwe_id:"294"})
MERGE (cwedata:Extended_Descriptions {runtime:253})
MERGE (cwe)-[:MAPPED_WITH]->(cwedata)
```

**Figure 19   Mapping between entities and CWE.**     

represent relationships between similar CVE entries, highlighting vulnerabilities that are being actively exploited by attackers.

### Final knowledge graph with deduction rule application

After storing the data in Neo4j, the KG was constructed with 155 nodes, 442 properties, and 175 relationships for CVE nodes. Additionally, we incorporated 6 CWE nodes, 36 properties, and 10 relationships into the KG. To further enrich the CWE descriptions and extend the graph, we linked specific CWE IDs to the extracted entities from extended descriptions, resulting in 174 nodes, 52 properties, and 25 relationships. By parsing the CPE values from CVE data, we created 33 CPE nodes, 573 properties, and 53 relationships. Furthermore, we added 88 nodes and 32 relationships based on the analysis of unstructured vulnerability description data.

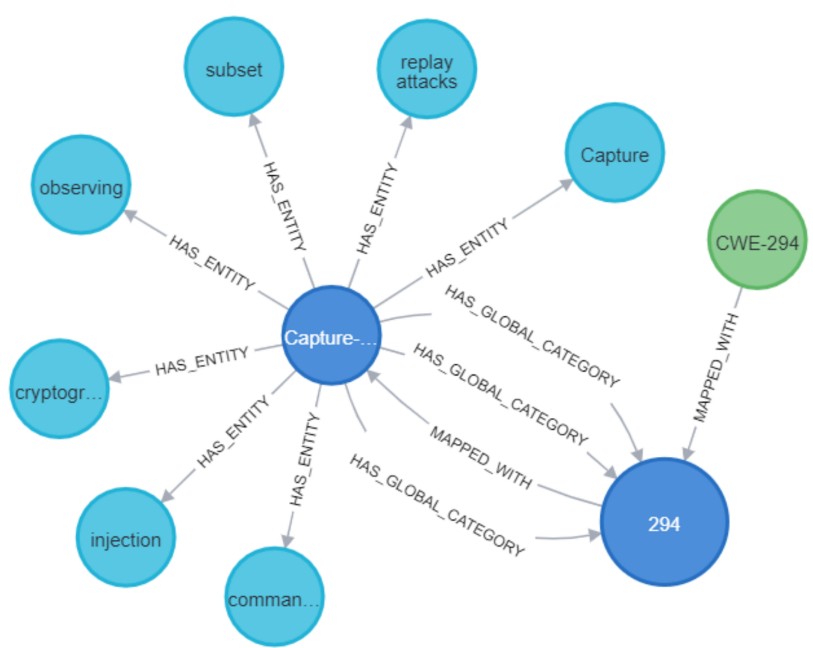

**Figure 20 The relationship between entities and CWE.**

MATCH p=(cve:CVE)-[:PROBLEM_TYPE]->(problemType:ProblemType)-[:MAPPED_WITH]->(cwe:CWE)

MERGE(cwe)-[:LEAD_TO]->(cve)

RETURN p

**Figure 21 Example of the first query.**

MATCH p=(cve:CVE)-[:PROBLEM_TYPE]->(problemType:ProblemType)-[:MAPPED_WITH]->(cwe:CWE)-
[:MAPPED_WITH]->(cwedata:Extended_Descriptions)-[:HAS_ENTITY]->(Entities:Entities{wikidataId:"Q2587068"})

MERGE(Entities)<-[:INADVERTENTLY_EXPOSES]-(cve)

MERGE (cve1:CVE{id:"CVE-2023-0012"})

MERGE (cve2:CVE{id:"CVE-2023-0017"})

MERGE (cve3:CVE{id:"CVE-2023-0023"})

MERGE (cve1)-[:SIMILAR_TO]->(cve2)

MERGE (cve2)-[:SIMILAR_TO]->(cve3)

RETURN p

**Figure 22 Example of the second query.**

Figures 23 and 24 display samples of the logical rules applied to KG. Figure 23 illustrates simple patterns where CWE leads to CVE vulnerabilities, while Fig. 24 shows complex patterns with similar relationships.

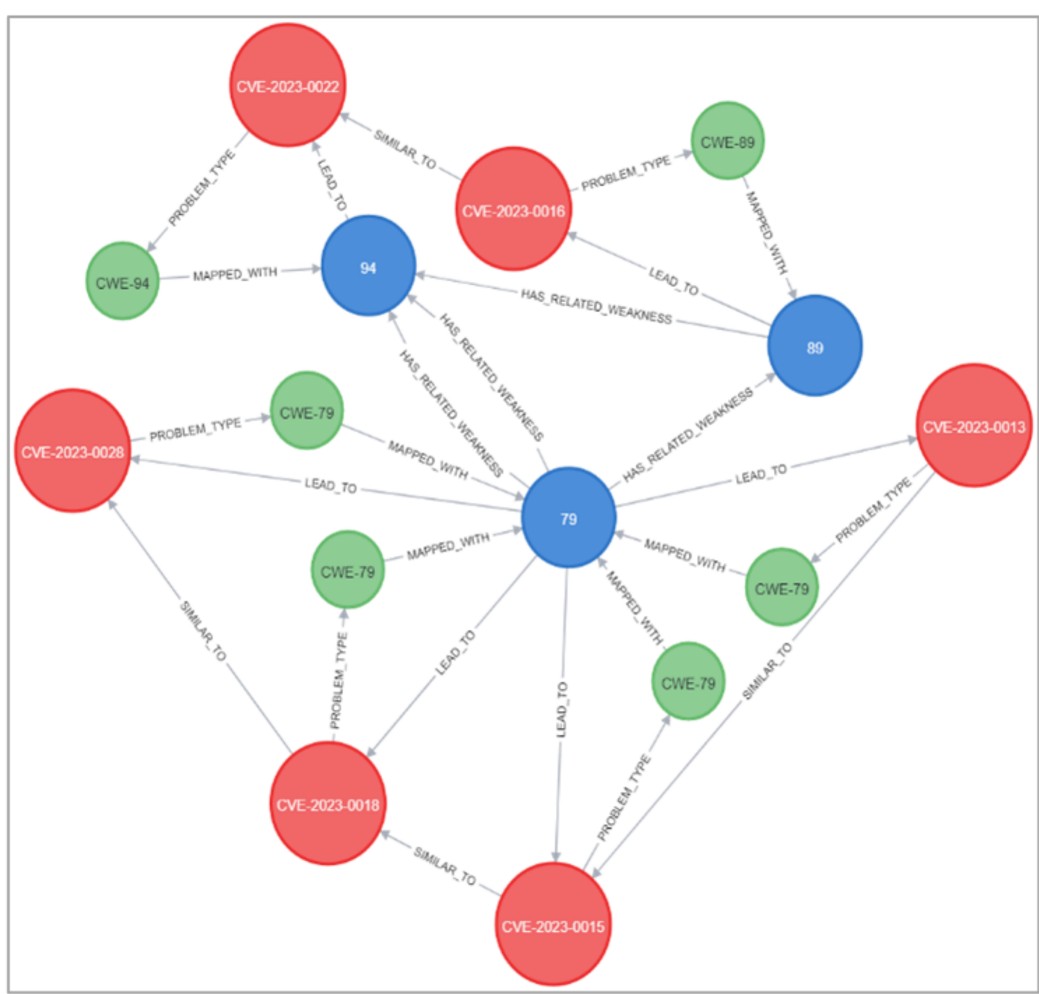

**Figure 23 The output of a sample of the logical rules from simple patterns.**

Figure 25 presents the final KG after importing and analyzing the heterogeneous data, followed by the application of logical rules to enhance the graph. Ultimately, our KG contains 612 entities and 758 relationships.

## Implementation of graph analytics algorithms

Graph-based analytics were implemented using the Neo4j Graph Data Science (GDS) library on the constructed KG. The datasets included in the analysis were accessible across all nodes and edges, providing a comprehensive view of the graph structure. Graph-based analytics involve a set of algorithms designed to model pairwise relationships among objects, which are represented as mathematical structures. These algorithms are particularly valuable in applications like semi-supervised learning, where they label unlabeled examples using limited labeled data combined with a large amount of unlabeled data. Additionally, in networks, they facilitate dimensionality reduction, clustering, and community detection. Graph-based analytics play a pivotal role in understanding the strength and direction of relationships between entities in a graph. This approach is

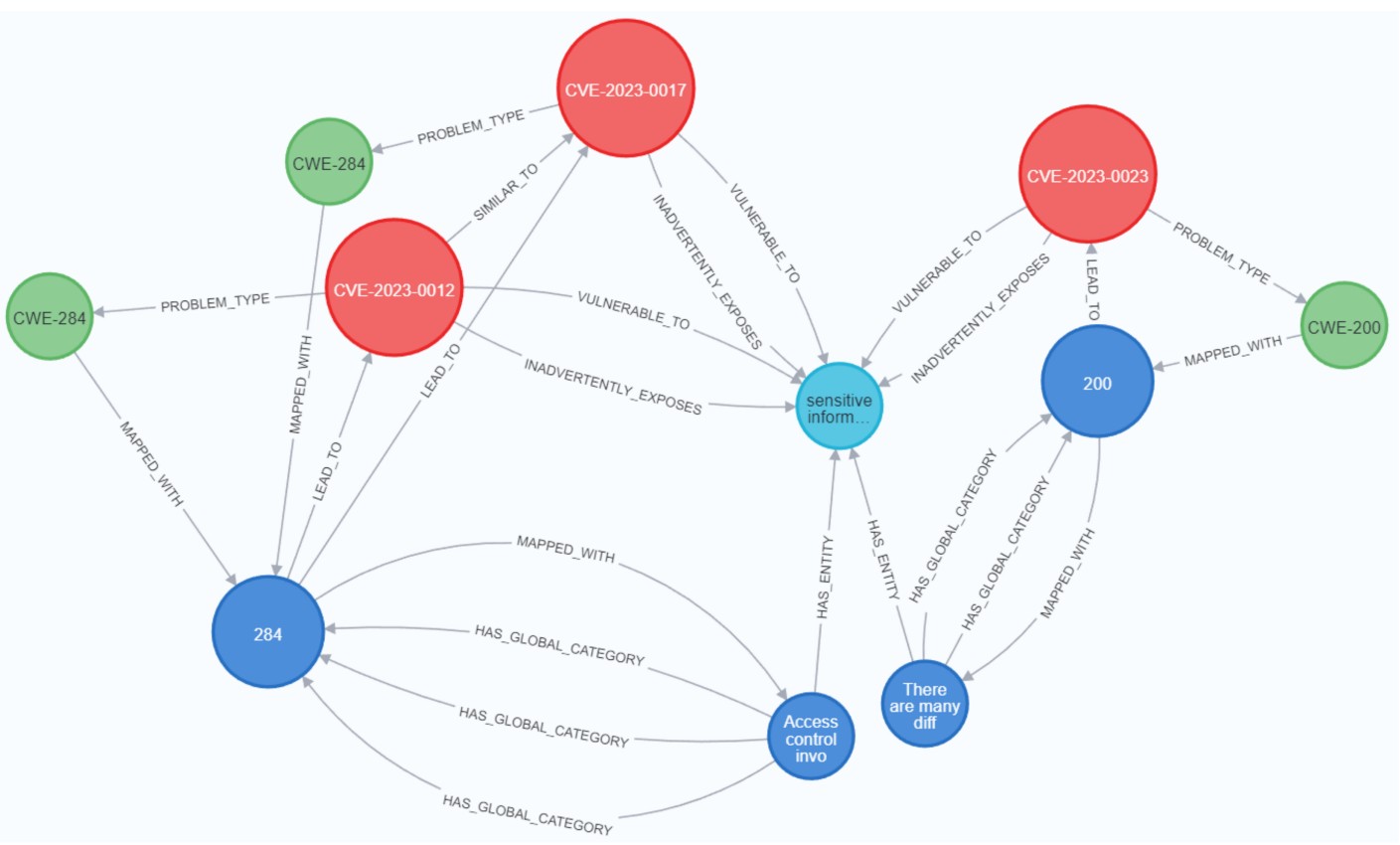

**Figure 24** The output of a sample of the logical rules from complex patterns.

particularly useful for unsupervised learning scenarios, allowing us to uncover patterns and relationships in the data. For example, the "importance" of a node can be evaluated similarly to assessing the relevance of a web page, where a page with many high-quality incoming links is considered more significant. This analogy highlights how relationships between nodes contribute to their importance.

We applied several graph analytics algorithms, focusing primarily on Centrality and Community Detection algorithms available in Neo4j Bloom, a visualization tool designed to explore and interact with graph data. The results of three algorithms measurement applied to the CWE subgraph are summarized in Table 3.

Centrality analysis

Centrality measures the importance of nodes within a network. Two centrality algorithms, Degree Centrality and PageRank, were applied:

1. Degree centrality: This algorithm measures the number of direct connections a node has. The results showed that the CWE node with ID 79 achieved the highest degree centrality score of 4, followed by CWE ID 284 with a score of 2. All other CWE nodes recorded a degree centrality score of 1.

PageRank: This algorithm identifies nodes with the most influence based on their connections and the quality of those connections. The analysis revealed that CWE 79 had the highest PageRank score of 1.26, establishing it as the most significant node in the
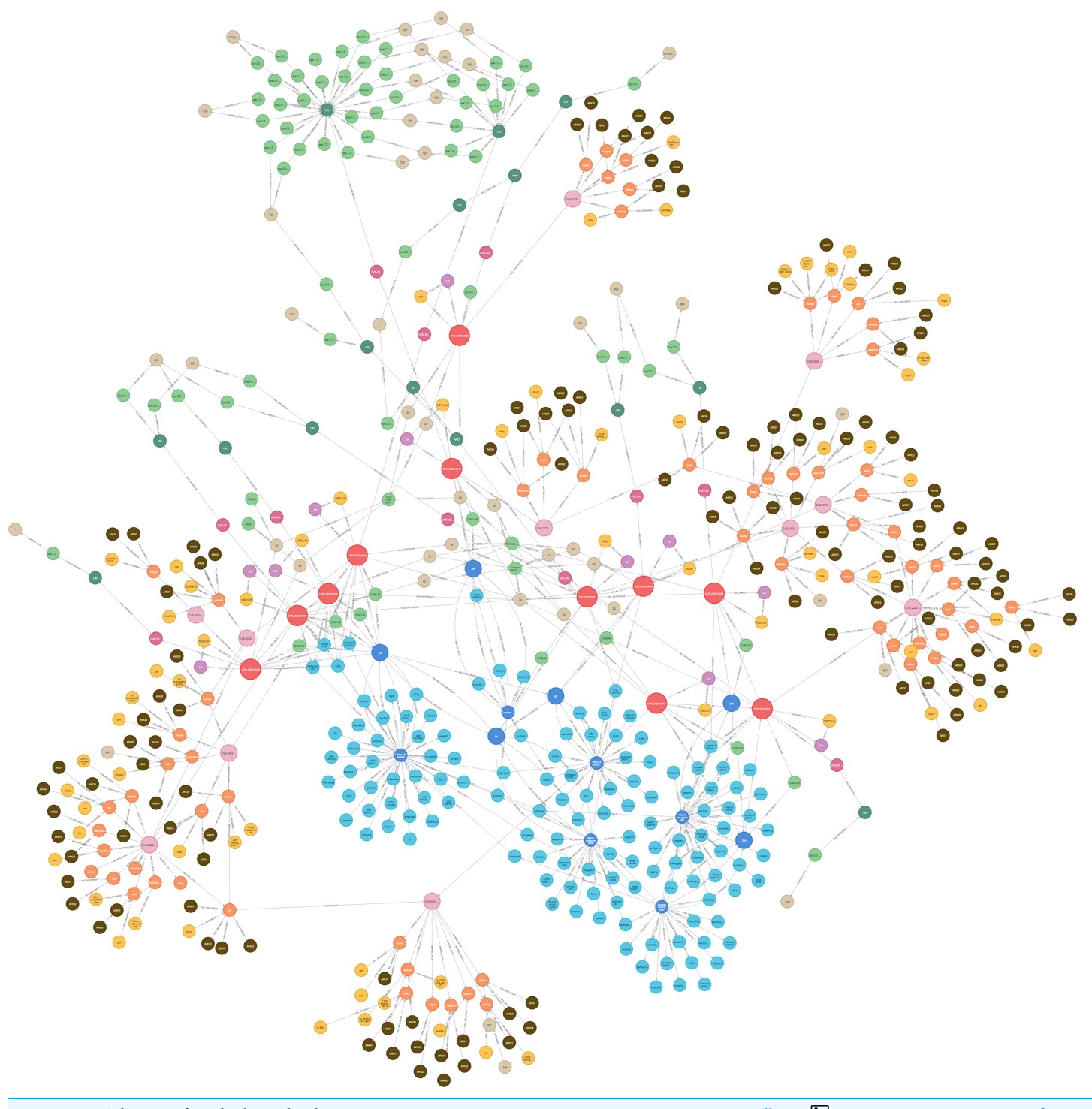

**Figure 25  The KG after the logical rules.**

network. This node is linked to four CVE vulnerabilities, contributing valuable insights to the Knowledge Graph. Notably, both centrality algorithms indicated CWE 79 as the most important node in the graph, as depicted in Fig. 26.

**Table 3  The results of three algorithms over KG.**

| Cwe Id | Degree centrality score | Page rank centrality score | Label propagation score |
|---|---|---|---|
| 79 | 4 | 1.264758277536877 | 155 |
| 200 | 1 | 0.4785000864294078 | 158 |
| 284 | 2 | 0.7405861501318974 | 159 |
| 89 | 1 | 0.4785000864294078 | 156 |
| 294 | 1 | 0.4785000864294078 | 160 |
| 94 | 1 | 0.4785000864294078 | 157 |

Community detection

To identify communities within the graph, we used the Label Propagation Algorithm (LPA). This algorithm groups nodes into communities by iteratively updating labels based on neighboring nodes. As shown in Fig. 27, nodes are color-coded according to their communities, with nodes in the same community sharing identical label propagation scores. However, it is important to note that the results may vary depending on the algorithm's configuration and the types of relationships and nodes analyzed.

The application of these graph analytics algorithms provided valuable insights into the structure and relationships within the knowledge graph, enabling a deeper understanding of the data and its inherent patterns.

Use cases for graph analytics

A graph analytics platform is essential for every organization that relies on connected data analysis to make critical decisions. These are some examples of graph analytics use cases. Fraud detection and analysis involves studying the interactions among different actors in a transaction. The purpose of this is to identify entities inside a system that are potentially troublesome and vulnerable to fraud. To prevent fraudulent behavior, it helps detect bad actors and implement countermeasures. It is also possible to use graph analytics to detect criminal activity and illegal behavior. Graph analysis is used by law enforcement for tracking phone calls, emails, people visiting suspects at specific locations, and monetary distribution networks in order to identify malignant and benign activity.

# EVALUATION

The objective of this section is to evaluate the construction and efficacy of the CKG by developing targeted queries and questions that showcase its capabilities. While the CKG is designed to address the broader challenges of cybersecurity, the current implementation focuses on vulnerability-related sources such as CVE and CWE. This focus serves as a proof of concept (PoC) to demonstrate the effectiveness of the proposed methodology for constructing and leveraging the CKG.

The purpose of this PoC is to validate the practical feasibility of integrating heterogeneous and complex cybersecurity data into a unified semantic framework. By harnessing structured and unstructured data, the CKG provides enriched semantic relationships and logical reasoning capabilities. Through this process, we bridge gaps in

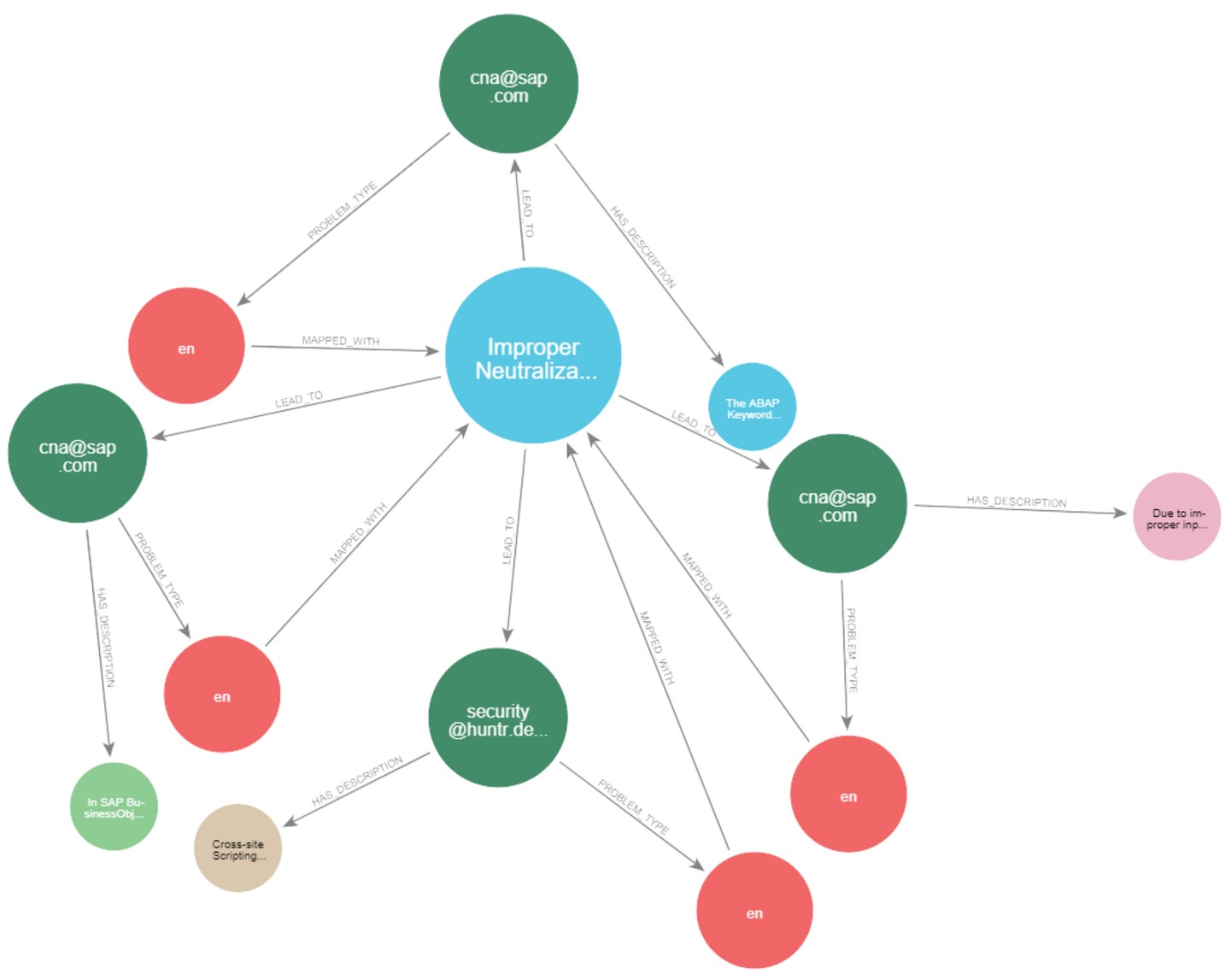

**Figure 26 The most important node is CWE 79.**

existing datasets, such as unstructured text and missing links between vulnerabilities and weaknesses, creating a coherent and machine-interpretable graph.

To evaluate the CKG, we devised a series of queries aimed at testing its ability to address challenges that traditional approaches cannot resolve. These queries validate the logical rules applied during the graph's construction and test its reasoning capabilities. For instance, the graph is designed to answer complex questions that involve implicit connections between entities, such as mapping specific weaknesses (CWEs) to their associated vulnerabilities (CVEs) or identifying potential attack vectors based on linked entities.

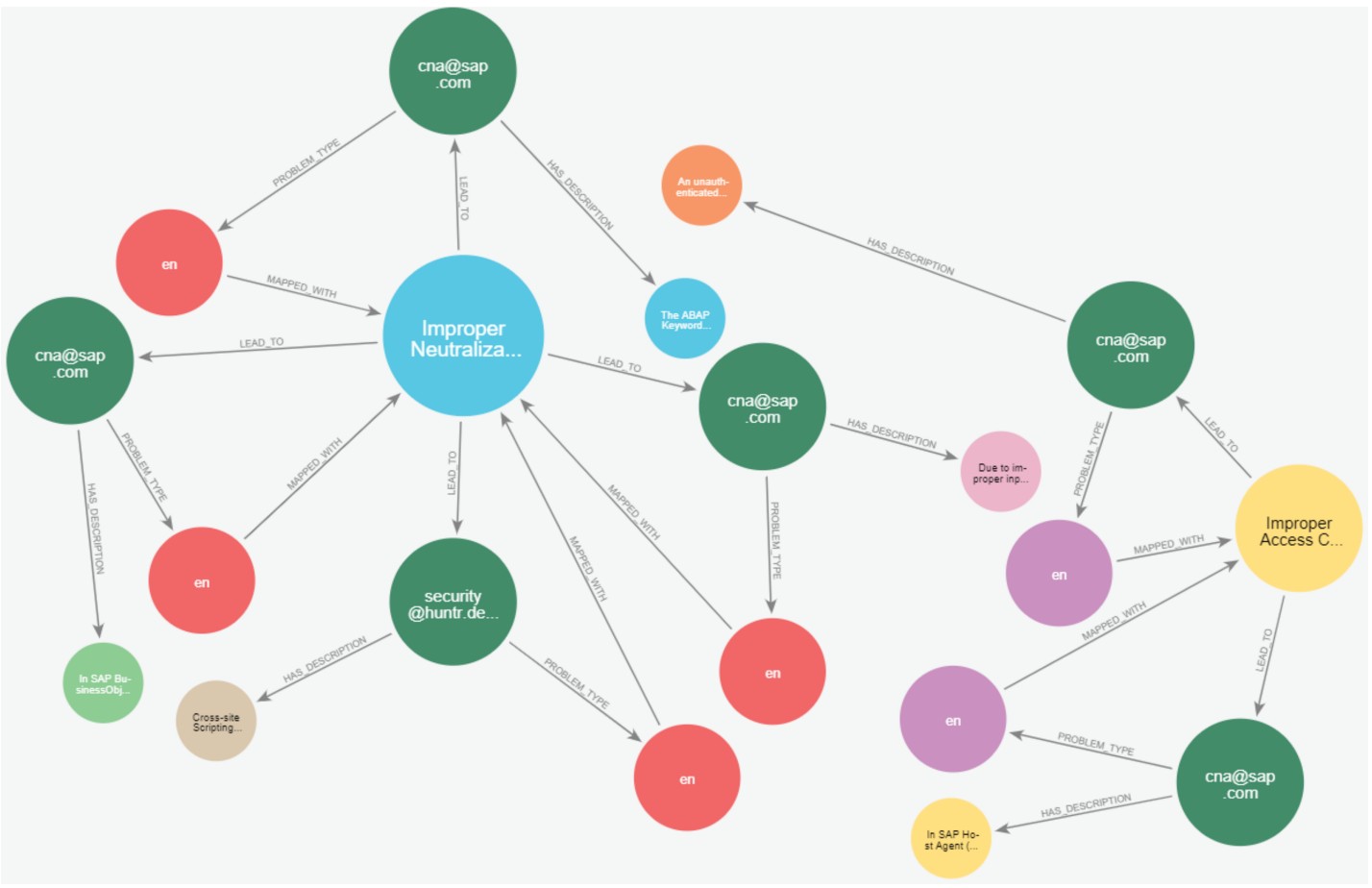

**Figure 27 A sample of the subgraph after using LP algorithm.** 

## Question answering-based evaluation

To evaluate the efficacy of our CKG, we demonstrated its ability to answer queries that could not be resolved using the original datasets in their raw JSON-based format, such as CVE and CWE datasets. This evaluation highlights the value of constructing the knowledge graph to establish meaningful connections between data entities and enable reasoning that is not possible with the original datasets alone.

### Querying the raw datasets

Initially, we queried the raw source files containing CVE and CWE data to illustrate their limitations. As shown in Figs. 28 and 29, data in the JSON-based format lacks explicit connectivity between entities, making certain types of questions unanswerable. For instance, when querying relationships between weaknesses (CWEs) and vulnerabilities (CVEs) using the query:

*MATCH (cwe:CWE) - [:LEAD_TO] -> (cve:CVE) RETURN cwe.cwe_id, cve.id*

No results were returned. This is because the necessary internal edges and relationships are not explicitly defined in the raw datasets. While internal edges within specific data items could be identified by analyzing the information provided in NVD documents, they

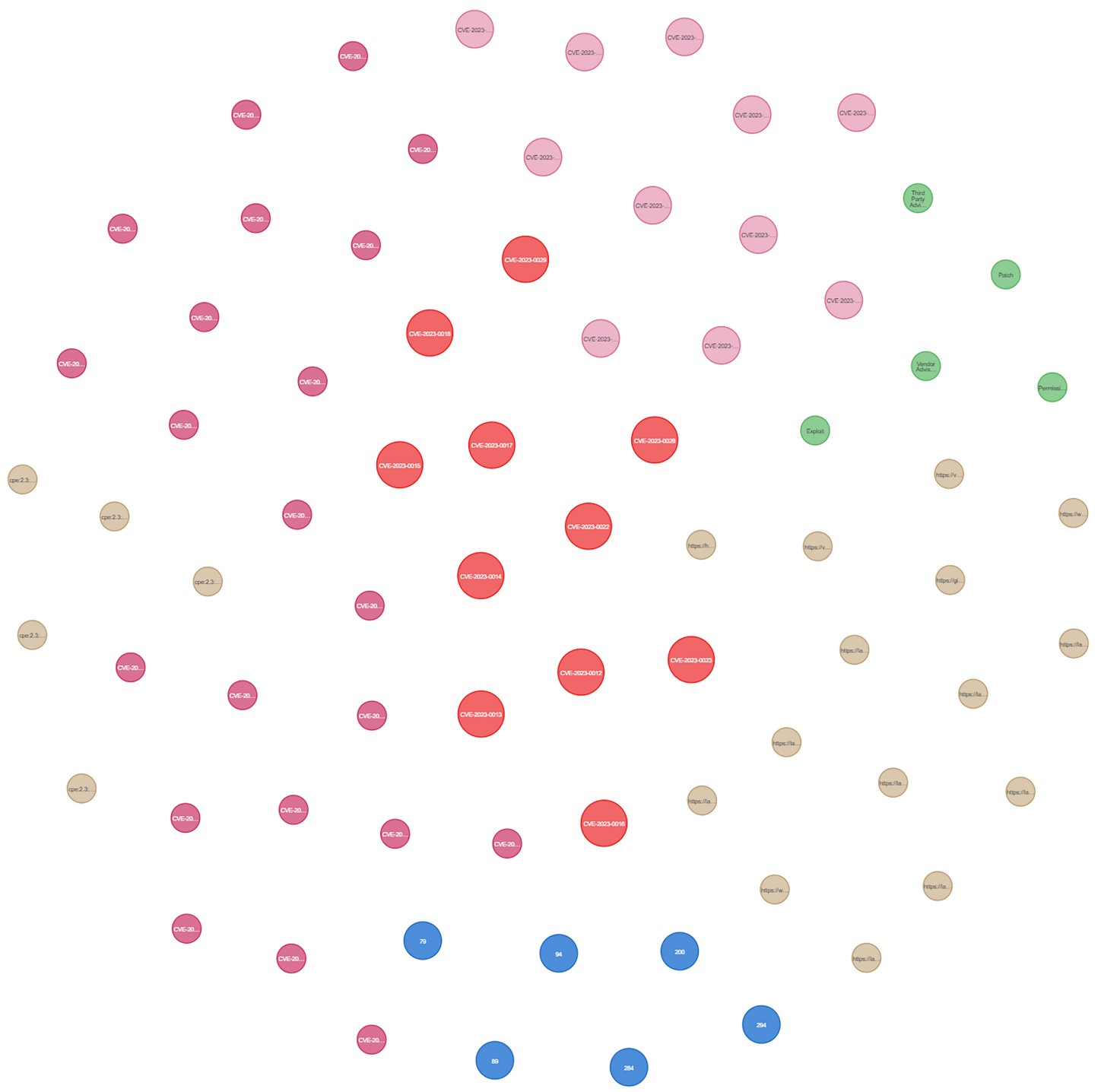

**Figure 28 Data not connected in the JSON-based model.**

```
1  MATCH (cwe:CWE)-[:LEAD_TO]→(cve:CVE)
2  RETURN cwe.cwe_id,cve.id
```

Table    (no changes, no records)

Code

**Figure 29 No results obtained from queries even after we created the internal edges inside a specific data item.**

**Table 4 Questions used to query the KG.**

| Question | Query |
|---|---|
| What are the weaknesses that lead to vulnerabilities? | MATCH (cwe)-[:LEAD_TO]->(cve) |
| | RETURN cwe, cve |
| What are the types of XSS attacks? | MATCH p=(cwedata:Extended_Descriptions)-[:HAS_ENTITY]->(Entities: Entities{wikidataId:"Q371199"}) |
| | RETURN p |
| What are the attacks an attacker can perform? | MATCH(E:Entities{wikidataId:"Q371199"}) |
| | MATCH(N:Entities{wikidataId:"Q228502"}) |
| | MATCH(T:Entities{wikidataId:"Q1756025"}) |
| | MATCH(I:Entities{wikidataId:"Q506059"}) |
| | RETURN E, N, T, I |
| What is the CVE ID that has no problemtype_data value? | MATCH (cve:CVE{id:"CVE-2023-0029"})-[:PROBLEM_TYPE]-> (problemType:ProblemType{value:"NVD-CWE-noinfo"}) |
| | RETURN cve, problemType |
| Which CVSS Score has the highest maximum and lowest minimum? | MATCH (cvssv3:CVSSV3) |
| | RETURN MAX(cvssv3.baseScore),MIN(cvssv3.baseScore) |
| How Can CVSS Severity and Score information, and the CVE ID and its value be sorted as a table or text? | MATCH (cvssv3:CVSSV3) WITH cvssv3 MATCH (description:Description) |
| | RETURN distinct (cvssv3.baseScore),(cvssv3.baseSeverity),(description. cveId), (description.value) |
| | ORDER BY cvssv3.baseScore, description.cveId |
| How many vulnerabilities are there in cna@sap.com? | MATCH (cve:CVE {assigner:"cna@sap.com"}) RETURN count(cve) |
| What are the vulnerabilities that can be exploited by attackers to perform attacks? | MATCH (Entities)<-[:VULNERABLE_TO]-(cve) |
| | RETURN cve, Entities |
| What are the CVSS scores that are assigned to vulnerabilities? | MATCH (cvssv3)-[:ASSIGNED_TO]->(cve) |
| | RETURN cvssv3, cve |
| What are the vulnerabilities that attackers can exploit by applying the same attack? | MATCH (cve1)-[:SIMILAR_TO]->(cve2) |
| | MATCH (Entities)<-[:VULNERABLE_TO]-(cve) |
| | RETURN cve1, cve2, Entities |
| What are the weaknesses that lead to vulnerabilities? | MATCH (cwe)-[:LEAD_TO]->(cve) |
| | RETURN cwe, cve |

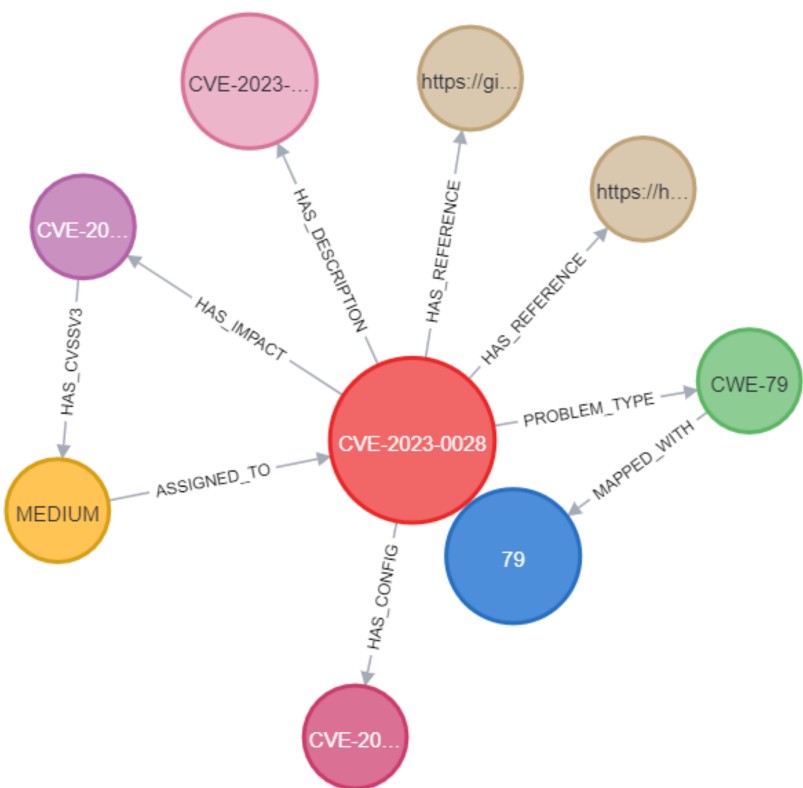

**Figure 30 The internal edges inside a specific data item, we created these edges after we analyzed the information provided by NVD documents.**

remained disconnected across datasets. This limitation demonstrates the need for a methodology to construct a cohesive KG.

### Knowledge graph-based querying

Using the proposed methodology, we constructed a CKG that enriches the raw datasets by establishing connections and applying logical rules are shown in Table 4. This allowed us to evaluate the CKG by enabling it to answer specific, complex questions. For example, by incorporating semantic relationships and logical reasoning, our CKG successfully addressed queries such as:

1. What are the weaknesses that lead to vulnerabilities?

• The query identifies weaknesses (CWEs) and connects them to vulnerabilities (CVEs) based on their relationships.

2. Which CVEs are associated with a specific CWE identifier?

• Each CVE entry in the CKG is linked to a corresponding CWE ID, enabling detailed analysis.

The ability to answer such questions demonstrates the CKG's capability to generalize patterns and validate connections established through logical rules.

### Validation of connections

The evaluation process involved validating the relationships (edges) in the CKG, which connect different data entities such as CVE and CWE nodes. These edges, derived from

logical rules, did not exist in the original JSON-based datasets. By querying both the JSON-based datasets and the CKG, we observed the following:

- JSON-based datasets: Queries failed to yield results due to the lack of explicit edges and connectivity, as shown in Fig. 28.
- CKG with logical rules: After applying the logical rules, the same queries returned meaningful results, demonstrating the effectiveness of the CKG refinement process.

### Results and impact

When the same queries were executed on the CKG, we obtained relevant data, as illustrated in Fig. 30. These results validate the improved semantic enrichment and reasoning capabilities enabled by the CKG. For instance, logical rules facilitated the discovery of implicit connections between CVEs and CWEs, allowing the CKG to answer questions that were previously unresolvable. By bridging the gaps in the raw datasets and enabling advanced querying, the CKG proves to be an invaluable tool for cybersecurity analysis, enhancing visibility and enabling experts to identify potential threats and vulnerabilities more effectively.

## CONCLUSIONS

Cybersecurity knowledge graphs offer significant potential to enhance online security and privacy by effectively organizing and linking crucial information about cyber threats and vulnerabilities. This research demonstrates the construction of an autonomous CKG from structured and unstructured sources, leveraging LPGs for efficient data representation. A key contribution lies in the development and implementation of a model for incorporating logical rules, which significantly improves the quality and completeness of the knowledge graph. Furthermore, the evaluation of the CKG using diverse graph analytics algorithms provides valuable insights into its performance and identifies areas for future improvement. While this research presents a valuable step forward in CKG development, challenges were encountered during implementation. Notably, the training of named entity recognition models proved to be particularly demanding, requiring substantial amounts of data for optimal performance. Given the limitations of existing pre-trained NER models in the cybersecurity domain, this research explored alternative approaches, including an entity-fishing tool and OIE from AllenNLP.

### Future research directions include

- Enriching the CKG: Integrating data from Microsoft Security Bulletins (MSB) and Cyber Threat Intelligence (CTI) reports will enable the inclusion of diverse attack patterns and scenarios, further enhancing the knowledge graph's value.
- Improving NER and OIE: Developing custom NER and OIE models specifically trained on cybersecurity data will significantly enhance the accuracy and completeness of entity and relation extraction.
- Advanced reasoning: Exploring and implementing more sophisticated reasoning methods, such as probabilistic reasoning and deep learning-based reasoning, will further improve the CKG's ability to infer new knowledge and detect complex relationships.

### Funding

This work was supported by the Deanship of Scientific Research, Vice Presidency for Graduate Studies and Scientific Research, King Faisal University, Saudi Arabia (Grant No. 5847). The funders had no role in study design, data collection and analysis, decision to publish, or preparation of the manuscript.

### Grant Disclosures

The following grant information was disclosed by the authors:
Deanship of Scientific Research, Vice Presidency for Graduate Studies and Scientific Research, King Faisal University, Saudi Arabia: 5847.

### Competing Interests

The authors declare that they have no competing interests.

### Author Contributions

- Hatoon Alharbi performed the experiments, analyzed the data, performed the computation work, prepared figures and/or tables, and approved the final draft.
- Ali Hur performed the experiments, analyzed the data, performed the computation work, authored or reviewed drafts of the article, and approved the final draft.
- Hasan Alkahtani conceived and designed the experiments, authored or reviewed drafts of the article, and approved the final draft.
- Hafiz Farooq Ahmad conceived and designed the experiments, analyzed the data, authored or reviewed drafts of the article, and approved the final draft.

### Data Availability

The code is available at GitHub and Zenodo:

- https://github.com/neostrange/cyberKG.

Alharbi, H. (2025). cyberKG Model. In PeerJ Computer Science (0.9). Zenodo. https://doi.org/10.5281/zenodo.14796080.

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
