# Peer review of "Enhancing cybersecurity through autonomous knowledge graph construction by integrating heterogeneous data sources"

_PeerJ Computer Science, doi:10.7717/peerj-cs.2768_

## Round 0.1 · original submission · Major Revisions

Dear authors,

Thank you for submitting your article. Feedback from the reviewers is now available. It is not recommended that your article be published in its current format. However, we strongly recommend that you address the issues raised by the reviewers, especially those related to readability, experimental design and validity, and resubmit your paper after the necessary changes and additions.

Best wishes,

Reviewer 1 ·

Basic reporting

The manuscript discusses enhancing cybersecurity capabilitites through constructing knolwedge graphs from heterogeneous data sources. The topic is highly actual and interesting.

The paper seems logically structured, but it is hard to evaluate due to placement of all the figures and tables outside the actual text. The review version is too long to grasp, I would prefer something closer to camera-ready in this case, but this is not author's fault, I assume. Also, section numbering would help a lot, I got lost easily in the manuscript, it is not clear what is section, what is subsection, etc.

Experimental design

Experiment design is fine.

A novel approach is proposed - does it have any name or can be referred to in other words than "proposed approach"?

Validity of the findings

Table 3 is actually one of the most interested parts of this work - the examples are fine, but did you consult them with cybersecurity professionals? It would be nice to show how KGs help in a specific step or workflow of a cybersecurity professional. This is a problem of the whole manuscript - what problems does it solve? It looks like another variation on related work, playing with NLP and KGs, but with no clear goal in mind. Otherwise, it is a nice paper, even though it is rather "mainstream" than "groundbreaking". It may be a nice contribution to the journal after a revision.

Additional comments

159 "However, there is still a lack of existing solutions that refine the KG, as most researchers did not refine the KGs and thus their KGs did not have consistency and accuracy." - how to understand this? Is the problem a lack of existing solutions or their low quality? What do you mean by consistency and accuracy? Can it be measured? Can autonomously constructed KGs be consistent and accurate?

The background sections lists some references, but the motivation section does not have any references - I would recommend using some, if possible, to streghten the statements.

Why have both background section and related work section? The first look like a brief version of the second one. They could be closer to each other, at least.

"Information extraction techniques" lists many related works and explains various aspects of them, not only information extraction techniques. Various metrics and results are mentioned - what are they? Please, add a section and explain various metrics used to evaluate the quality of KGs and reasoning on them - this is critical and not covered in related work.

Did you consider processing other data than NVD? It is widely-known source of textual cybersecurity data, but the whole field seems to process only those. What alternatives are there and what could be achieved with them?

863 "The use of graph analytics for national security and defense is controversial because it compromises the privacy of citizens." - what? This depends on the data used, not the method of their processing.

Reviewer 2 ·

Basic reporting

- The English language should be improved so that an international audience can clearly understand your text. The current phrasing makes comprehension difficult. Some examples are
- l35: it's -> it has
- l38: Where small
- l51: there are different sources -> different source
- l90: "lack of language understanding" -> vague information
- l99: different types of formats -> different formats
- l104: Cite LPG
- l131: KG "are" highly useful
- l146: which types .... be tackled
- l160: did not have consistency or accuracy -> vague (clarify and cite)
- l171-174: many previous works mentioned but not cited
- l186: cite previous works
- l188: cite previous works
- l203: unclear whether it is 95% or 80% -> rephrase
- l221: high: vague (quantify)
- l234: Reasoning Function: vague (clarify)
- l469: named??
- l535: John managed to open -> John managed to open the door
- l682,687: Use the same representation of spaCy or Spacy
- l689: Proper names or proper nouns
- l713: CWE documents mentioned twice
- l763: somewhat long sentences -> vague (rephrase or give statistics)
- l817: manually examined -> not clear what this means
- l838: is it two or three, the table shows three measures
- l892 these lists cannot be directed answered -> clarify which lists
- l894: run the questions -> answer instead of run?
- l910: Json-based model -> model or dataset
- l912: edges not existed -> do not exist

- The Introduction and background are well written and show enough relevant articles which build up the context for the research work
- The structure conforms to the standards
- Figures are relevant and helped me understand the texts better. However, in some figures, it is hard to see the contents of the nodes of the graph (for eg, in Fig 25, relations in Fig 26,27, and Nodes in Fig 28). A table containing all node names should be provided and then referenced to the image

Experimental design

- The design of experiments and research questions are relevant and well-defined.
- Multiple experiments and investigations performed which validate the claims
- Most of the methods are clear. I have some clarifying questions which should be addressed in the revision to make the approach clear.
- A table for the different types of data should be provided for readers to have a clear understanding of the heterogeneous data sources. I suggest having a table with fields like source (dataset name), nature (structured or unstructured or both), volume of data points, fields (authors used or present in the data if structured))
- l545 -> What patterns created? Please clarify in detail
- For sections f (l550), g(l560), h(l574), i(l585), and j(l598) use examples from your data to illustrate instead or using general natural language examples so that the readers have clear understanding on the data used.
- It is not clear to me how the disambiguation is done since the entities usually extracted by pre-trained models are different that specific cyber-security entities.
- I suggest giving a table for evaluation (Line 871) on how many questions asked and how many answered to qualify the results. Use accuracy or other metrics to measure

Validity of the findings

- The overall approach is novel and has a potential impact, especially the KG extension part
- The created CKG will be helpful for future cyber-security
- The conclusion is well stated supporting the results and linked to the original research questions

---

## Round 0.2 · accepted · Accept

Dear Author,

I would like to express my gratitude for the revised paper. It is worth noting that one of the reviewers did not respond to the invitation to review the revision. However, one reviewer has accepted the paper. I have also conducted my own assessment of the revision. The paper appears to have undergone sufficient improvement, and I am content with the current version. Following the rectification of minor errors pertaining to the alignment and writing style, the revised paper manuscript appears to be prepared for publication.

Best wishes,

Reviewer 2 ·

Basic reporting

Thank you authors for going through my comments and questions and answering them in the rebuttal

* The authors have improved their writing and corrected the earlier-mentioned typos and minor presentation issues.
* For the clarifying questions asked to the authors, the rebuttal documents clearly explained the answers.
* The authors also incorporated the suggestions in the new version of the paper
* They also improved their illustrations, presentation, and image qualities which improved the overall content.

I am satisfied with the changes made.

Experimental design

.

Validity of the findings

As mentioned in my prior review the findings are valid and interesting backed with experimental results.